# Radiative Heating Rate Profiles over the Southeast Atlantic Ocean during the 2016 and 2017 Biomass Burning Seasons

Allison B. Marquardt Collow[1,2], Mark A. Miller[3], Lynne C. Trabachino[4], Michael P. Jensen[5], and Meng Wang[5]

[1]Universities Space Research Association, Columbia, Maryland, USA
[2]Global Modeling and Assimilation Office, NASA Goddard Space Flight Center, Greenbelt, Maryland, USA
[3]Department of Environmental Sciences, Rutgers University, New Brunswick, New Jersey, USA
[4]Institute for Earth, Ocean, and Atmospheric Sciences, Rutgers University, New Brunswick, New Jersey, USA
[5]Environmental and Climate Sciences Department, Brookhaven National Laboratory, Upton, New York, USA

*Correspondence to*: Allison B. Marquardt Collow (allison.collow@nasa.gov)

**Abstract.** Marine boundary layer clouds, including the transition from stratocumulus to cumulus, are poorly represented in numerical weather prediction and general circulation models. Further uncertainties in the cloud structure arise in the presence of biomass burning carbonaceous aerosol, as is the case over the southeast Atlantic Ocean where biomass burning aerosol is transported from the African continent. As the aerosol plume progresses across the southeast Atlantic Ocean, radiative heating within the aerosol layer has the potential to alter the thermodynamic environment and therefore the cloud structure; however, limited work has been done to quantify this along the trajectory of the aerosol plume in the region. The deployment of the First Atmospheric Radiation Measurement Mobile Facility in support of the Layered Atlantic Smoke Interactions with Clouds field campaign provided a unique opportunity to collect observations of cloud and aerosol properties during two consecutive biomass burning seasons during July through October of 2016 and 2017 over Ascension Island (7.96 °S, 14.35 °W). Using observed profiles of temperature, humidity, and clouds from the field campaign, alongside aerosol optical properties from MERRA-2 as input for the Rapid Radiation Transfer Model, profiles of the radiative heating rate due to aerosols and clouds were computed. Radiative heating is also assessed across the southeast Atlantic Ocean using an ensemble of back trajectories from the Hybrid Single Particle Lagrangian Integrated Trajectory Model. Idealized experiments using the Rapid Radiation Transfer Model with and without aerosols and a range of values for the single scattering albedo demonstrate that shortwave heating within the aerosol layer above Ascension Island can locally range between 2 and 8 K per day depending on the aerosol optical properties, though impacts of the aerosol can be felt elsewhere in the atmospheric column. When considered under clear conditions, the aerosol has a cooling effect at the TOA and based on the observed cloud properties at Ascension Island, the cloud albedo is not large enough to overcome this. Shortwave radiative heating due to biomass burning aerosol is not balanced by additional longwave cooling, and the net radiative impact results in a stabilization of the lower troposphere. However, these results are extremely sensitive to the single scattering albedo assumptions in models.

# 1 Introduction

Marine stratocumulus and trade wind cumulus are prominent cloud types over the Atlantic Ocean with regional and global impacts on the energy budget (Bony and Dufresne, 2005). Despite their importance, models struggle to accurately represent these clouds and their properties. Within the southeast Atlantic and other subsidence regions, general circulation models and reanalyses tend to underestimate the cloud fraction (Klein et al., 2013; Dolinar et al., 2015) and optical thickness of warm marine stratocumulus (Lin et al., 2014; Noda and Satoh, 2014; Rapp. 2015). Furthermore, models struggle to properly link environmental conditions to cloud properties of trade wind cumuli (Nuijens et al., 2015). The uncertainty and discrepancy among models within the region are further complicated by the presence of biomass burning aerosol (Stier et al., 2013; Peers et al., 2016). Using global model simulations, it was shown by Brown et al. (2018) that the largest radiative impact from brown carbon occurs off the west coast of southern Africa. Biomass burning aerosol that gets entrained into marine stratocumulus in the southeast Atlantic has a larger impact on the radiation budget than the direct radiative effect of the aerosol itself (Lu et al., 2018). The determination to answer questions and resolve uncertainties surrounding this topic in the southeast Atlantic Ocean led to an international effort termed COLOCATE (Clarify-Oracles-Lasic-aerOClo-seAls Team Experiment), with overlapping field campaigns and modeling studies from the United Kingdom, France, South Africa, Namibia, and the United States (Zuidema et al., 2016). The focus here is a combination of radiation transfer modeling and observations from DOE's Layered Atlantic Smoke Interactions with Clouds (LASIC) campaign.

Originating in the savannas of southwestern Africa, biomass burning aerosol plumes extend up to between 3.5 to 4.5 km above ground level and is transported via the Southern Africa Easterly Jet over the southeast Atlantic Ocean where the aerosol plume begins to descend (Adebiyi et al., 2015; Adebiyi and Zuidema, 2016; Das et al., 2017). Fires and the associated aerosol in this region are typical during the months of July through October. When compared to satellite observations, global models commonly simulate that the biomass burning aerosol descends too rapidly once over the ocean (Das et al., 2017; Gordon et al., 2018), which can have implications on the thermodynamic structure and can indicate dynamical deficiencies. While over the ocean, initial space-based observations indicate that the aerosol plume is primarily above the boundary layer. Over Ascension Island, a remote island located roughly 1600 km from the African continent, the aerosol tends to be in the boundary layer during the beginning of the biomass burning season but is located above the cloud layer towards the end in September and October (Zuidema et al., 2018b).

Biomass burning aerosol in the Southeast Atlantic region and its impact on heating within the column has been investigated through recent modeling experiments (Chang and Christopher, 2017; Lu et al., 2017, Gordon et al., 2018; Mallet et al., 2019). Heating rate profiles within the region were calculated by Chang and Christopher (2017) using the Santa Barbara DISORT Atmospheric Radiative Transfer (SBDART) model and fixed values for aerosol and cloud properties corresponding to Southern AFricAn Regional science Initiative (SAFARI 2000) observations. Chang and Christopher (2017) noted that with fixed aerosol and cloud properties, the radiative heating rate increased throughout the biomass burning season due to the

decreasing solar zenith angle. This study also determined the solar zenith angle (54°) at which the direct radiative effect of aerosols located above liquid clouds is maximized. Lu et al. (2017) used large eddy simulations nested within Weather Research and Forecasting with Chemistry (WRF-Chem) to quantify the microphysical, direct, and semi-direct effects of aerosol within the Southeast Atlantic.  A total cooling of roughly 8 W m$^{-2}$ in the shortwave (SW) was found at the top of the atmosphere with a large component of that from the microphyscial effects of biomass burning aerosols on clouds as a result of the Twomey effect, higher liquid water path, and higher cloud fraction before noon (Lu et al., 2017). Another recent study by Gordon et al. (2018) quantified radiative heating within the atmospheric column by switching biomass burning aerosols and absorption due to biomass burning aerosols on and off in a hybrid of the regional configuration of the UK Chemistry and Aerosol Model and HadGEM. While Gordon et al. (2018) established the use of the hybrid model combination for aerosol studies and identified discrepancies between the model and observations, only the first ten days of August 2016 were analyzed.

The primary goal of this study is to quantify the individual impact of clouds, black carbon individually, and all aerosols collectively on heating within the atmospheric column above Ascension Island in the Southeast Atlantic, as well as the uncertainty that exists in the radiative heating rates. Radiative heating due to aerosol within the cloud layer has long been hypothesized to alter the thermodynamic profile, stabilize the boundary layer, and suppress convection in trade cumulus (Ackerman et al., 2000). However, the opposite effect can be true when the aerosol is located above the cloud layer, resulting in an increase in cloudiness (Johnson et al., 2004, Adebiyi, 2016). An added complication to this radiative heating due to aerosol in the Southeast Atlantic arises from uncertainties associated with the aerosol optical properties. Not only do models produce a range of values for the single scattering albedo (SSA) with different wavelength dependencies but observed values for the SSA can vary within the region depending on the instrument used (Pistone et al., 2019; Shinozuka et al., 2019).  Previous studies of the radiative heating rate within the column in the Southeast Atlantic are expanded upon by employing varying thermodynamic, cloud, and aerosol properties using ground-based observations and observation-constrained aerosol profiles from reanalysis throughout the biomass burning seasons of 2016 and 2017 over Ascension Island. Our approach uses these observations and analyses of aerosol and cloud properties as input to a radiative transfer model to produce a possible range of heating rates associated with uncertainties in the SSA and back trajectories of the aerosol plume as it is transported across the Southeast Atlantic. This in turn can be used to determine how the thermodynamic profile is altered by aerosols, and the resulting modifications to the formation and maintenance of clouds in response to this heating. A Lagrangian approach for the region, such as this, was recommended by Diamond et al. (2018).

Section 2 describes the observational and reanalysis data sets that are used in this study as well as the methodology for idealized radiation transfer simulations. An evaluation of aerosol optical depth (AOD) and the vertical profile of aerosols in the reanalysis product is presented in Section 3 and Section 4 discusses thermodynamic profiles of temperature, relative

humidity, and cloud microphysical properties. Results of the radiative heating rates due to atmospheric constituents are detailed in Section 5, while a discussion and conclusions can be found in Section 6.

## 2 Data and Methodology

### 2.1 ARM Mobile Facility and Value-Added Products

Observations of thermodynamic profiles, clouds, and aerosols used in this study are from the First Atmospheric Radiation Measurement (ARM; Mather and Voyles, 2013) Mobile Facility (AMF1; Miller et al., 2016), which was located on Ascension Island (7.7 °S, 14.35 °W, 340.77 m) from 1 June 2016 through 31 October 2017 with the objective of observing two consecutive biomass burning seasons. While the AMF1 was stationed on the windward side of the island, radiosondes were launched at the airport on the southeastern side of the island near an existing Aerosol Robotic Network (AERONET) site

(Zuidema et al., 2018a). The interpolated sounding (INTERPSONDE) value-added product (VAP) is used for temperature and humidity profiles (ARM Climate Research Facility, 2016a; ARM Climate Research Facility 2016b). INTERPSONDE is anchored by six hourly radiosonde launches and a linear interpolation is used to fill in time steps between launches (Toto and Jensen, 2016). Evidence of ground check artefacts were present in the radiosonde data and were not fixed prior to the interpolation. These artefacts have been removed as part of our postprocessing. Microwave radiometer retrievals (MWRRET;

Gaustad et al. 2011) of precipitable water vapor are used to further constrain the humidity profiles. The resulting INTERPSONDE data has a temporal resolution of one minute and vertical resolution ranging from 20 m to 500 m, depending on the height above ground level. Aerosol optical depth (AOD) was observed using a multifilter rotating shadowband radiometer (MFRSR) and calculated using the 1st Michalsky algorithm (Koontz et al., 2013). Additional measurements of AOD from AERONET were taken using a Cimel sun photometer (Holben et al., 2001; Giles et al., 2019).

Cloud properties used in the radiation transfer simulations were determined using a Ka-band cloud radar, micropulse lidar, and laser ceilometer, with the data combined into the Active Remote Sensing of Clouds (ARSCL) VAP at a temporal resolution of four seconds (Clothiaux et al., 2000). Cloud properties including cloud liquid/ice water content and liquid/ice cloud droplet effective radius were determined using the method presented in Dunn et al. (2011) and currently used in MICROBASE, which is a retrieval algorithm utilizing constrained data from ARSCL as well as the microwave radiometer and

INTERPSONDE profiles. The accuracy of this retrieval algorithm has been evaluated using radiative closure experiments and it is known to be accurate enough to adequately represent radiation transfer through clouds. It has been used in past studies (eg. Mather et al., 2007) to estimate tropical heating rate profiles. Complete validation of such an algorithm is not possible using in-situ measurements, but its reliance upon cloud liquid water path and its use in the tropical atmosphere are consistent with its capability. All clouds that are colder than -16° C were considered to be comprised entirely of ice, while all clouds

above 0° C were liquid. A linear fractionation scheme was used to partition particle phase in the region between 0 and 16° C. It is worth noting however that clouds over Ascension Island are primarily liquid. Thorough comparisons to other retrieval

algorithms, and evaluations of the relative performance of the MICROBASE algorithm are presented by Zhao et al. (2012) and Huang et al. (2012).

## 2.2 MERRA-2

The vertical profile of aerosols and their column integrated properties can be difficult to continuously observe, especially during cloudy conditions. Throughout the LASIC campaign, there were numerous hours without observations of AOD. In order to maximize time steps when heating rate profiles could be calculated given the near constant partly cloudy skies over Ascension Island, aerosol properties from the Modern-Era Retrospective analysis for Research and Applications, Version 2 (MERRA-2; Gelaro et al., 2017; GMAO, 2015a; GMAO, 2015b) were instead used for the radiation transfer simulations. MERRA-2 is the latest contemporary reanalysis from NASA that has the advantage of assimilated AOD, a feature that is not present in other reanalysis products. The decision to use MERRA-2 was made such that we would have a self-consistent data source of aerosols, clouds, and thermodynamic profiles to use for heating rate profiles along the back trajectory of the aerosol plume as it is transported from southern Africa to Ascension Island. Cloud and thermodynamic profiles from MERRA-2 were only used in the radiation transfer calculations along the back trajectory discussed in Section 5.4. MERRA-2 data are available at a spatial resolution of roughly 50 km and 72 vertical levels from the surface through 0.1 hPa and a temporal resolution of one hour for single level variables and three hours for three dimensional variables.

The dominant observational source of AOD that is assimilated into MERRA-2 is Collection 5 bias-corrected Moderate Resolution Imaging Spectroradiometer (MODIS) AOD (Randles et al., 2017). Other aerosol datasets are assimilated into MERRA-2 however they are not applicable for the time period of the LASIC campaign. Daily emissions of biomass burning aerosol come from the Quick Fire Emissions Dataset (QFED) version 2.4-r6 (Darmenov and da Silva, 2015). Within MERRA-2, aerosols are simulated using the Goddard Chemistry, Aerosol, Radiation, and Transport Model (GOCART), which separates the AOD into five species, sea salt, dust, sulfate, organic carbon, and black carbon, and defines the vertical distribution of aerosols. Further details on the assimilation of aerosols in MERRA-2 can be found in Randles et al. (2017), while an evaluation with respect to independent observations can be found in Buchard et al. (2017). MERRA-2 aerosols during the LASIC campaign are further evaluated in Section 3.

## 2.3 Rapid Radiative Transfer Model

The Rapid Radiative Transfer Model (RRTM) was used to perform idealized experiments to calculate the SW heating within the column due to black carbon, all aerosols, and clouds. A user-specified vertical profile was used with the temperature and humidity profiles coming from INTERPSONDE and cloud properties from MICROBASE. Prior to insertion into RRTM, the INTERPSONDE profiles were interpolated onto the MERRA-2 vertical levels based on height above ground level to match the resolution of the aerosol vertical profiles. RRTM runs were performed every four seconds to match the temporal resolution of MICROBASE, while solar zenith angle was updated every fifteen minutes, the temperature and humidity profiles hourly, and aerosols every three hours due to the temporal resolution of vertical profiles in MERRA-2. Aerosol optical properties,

including AOD, angstrom exponent, and SSA are from MERRA-2, and were scaled in the vertical by the profile of mixing ratio for the individual species (GMAO, 2015; GMAO, 2015b). The value for SSA at 550 nm from MERRA-2 was used and assumed to be spectrally independent, with average values during the month of August 2016 of 0.99 just above the surface, decreasing to roughly 0.93 within the aerosol layer. Two variations upon the MERRA-2 SSA were also used to account for potential deficiencies in the humidity profile and aerosol speciation, allowing the monthly mean SSA to drop to ~0.91 for the humidity correction and ~0.82 when the SSA is reduced for organic carbon. Asymmetry parameter was assumed to be 0.756, the value given by Hess et al. (1998) for a polluted maritime air mass. Other values of asymmetry parameter were tested but did not impact the results. A total of six sets of experiments were completed to quantify the individual and combined contribution of clouds and aerosols: 1) Clean and clear sky without clouds or aerosols, 2) Clear sky with all aerosols, 3) Clear sky with all aerosols except black carbon, 4) Clean and cloudy sky, 5) Cloudy sky with all aerosols, and 6) Cloudy sky with all aerosols except black carbon. A summary of the experiments and the fields they were used to calculate can be found in Tables 1 and 2, respectively. All six experiments were repeated using three different values for SSA as described in Section 5.1. Clear sky simulations were performed by turning off clouds in the radiation transfer model. This means that the radiation transfer may still feel the impact of clouds through the enhanced humidity in the thermodynamic profiles. The impact of this on the results is likely small given the same thermodynamic profiles are used for all experiments. It is worth noting that a true assessment of heating due to biomass burning aerosol should isolate brown carbon, however that is not an aerosol species available in MERRA-2 at this time.

## 2.4 Back Trajectories

Optical properties and radiative effects of aerosols are dependent on their location with respect to clouds, as well as the solar zenith angle (Chang and Christopher, 2017). As a result, the radiative impact of biomass burning aerosol, and therefore its impact on the thermodynamic profile and clouds prior to reaching Ascension Island, is dependent on the back trajectory of the aerosol plume. To determine the path of the aerosol plume and how it differs between the 2016 and 2017 biomass burning seasons, the HYbrid Single Particle Lagrangian Integrated Trajectory (HYSPLIT) model was used to compute ten day back trajectories for a parcel originating at Ascension Island at 12z on each day in August and September 2016 and 2017, driven by the large scale meteorology from MERRA-2 (Stein et al., 2015). Based on results for the height of the aerosol plume (Figures 2 and 3, Figure 4 of Zuidema et al., 2018), the parcel originated at a height of 2 km. An additional set of back trajectories were calculated in an identical manner for a case study originating at Ascension Island on 13 August 2016 using input from the 27 ensemble members of NCEP's Global Data Assimilation System (GDAS) at 0.5-degree spatial resolution.

## 3 Evaluation of Aerosols in MERRA-2

Previous evaluations of aerosol properties in MERRA-2 have been limited so it is therefore essential to ensure that MERRA-2 is representative of the observations that are available from the AMF1 when it was stationed in Ascension Island.

Aside from observations from the AMF1, there are also AOD observations from an existing AERONET site located near the airport where the soundings were launched (Holben et al., 2001). Daily mean AOD from the two observational sources as well as from MERRA-2 for August and September 2016 and 2017 can be seen in Figure 1. It can readily be seen that observations from the AMF1 are limited in all four months due to cloudiness over the site. Therefore, correlation coefficients and biases

presented in Figure 1 were calculated for MERRA-2 with respect to AERONET observations only including days when observations were available. When it was cloudy, AERONET was not able to measure AOD. The highest aerosol loading over Ascension Island was present in the middle and end of August 2016, with daily values of AOD ranging from 0.1 the first couple days of the month to a maximum of 0.73 on 13 August 2016, followed by additional periods of elevated AOD during September 2017. These values for AOD are similar to those presented by Zuidema et al. (2016) using AERONET observations

over the period of 2000 through 2013. A periodicity can be seen in each of the four months as the aerosol plume drifts overhead of Ascension Island. Correlations between AOD in the observations and MERRA-2 exceed 0.8 in all four months. The largest bias of 0.04 with respect to AERONET occurs in August 2017 however the AERONET observations are also generally higher than those from the AMF1.

          Observations of SSA during LASIC were presented by Zuidema et al. (2018b) and monthly mean values of 0.78 and

0.81 were specified during August and September at a wavelength of 529 nm. When all of the aerosol species are considered in MERRA-2, the SSA tends to be a bit higher, with monthly mean values of 0.92 and 0.93 for August and September respectively. There are a few possible explanations for this discrepancy. In reality, much of the organic biomass burning aerosol can be considered brown carbon, a species that is not represented in GOCART and the Goddard Earth Observing System (GEOS), the underlying model and data assimilation system in MERRA-2. Brown carbon tends to be more absorbing than

organic carbon and therefore if included, the SSA could be lower. In addition, the optical properties for aerosols in MERRA-2 are defined by a look up table as a function of relative humidity. Differences in the thermodynamic profile will therefore result in a different SSA. An additional concern is that the observations are representative of the aerosol within the boundary layer, while values given for MERRA-2 consider the entire column. Additionally, differences could stem from limitations of the nephelometer, which only allowed for a relative humidity between 45 and 65% and a particle size less that 1 micron. This

means that scatter due to larger particles such as sea salt is not represented in the observed value for the SSA. On the contrary, the SSA in MERRA-2 is more aligned with those presented for the region by Pistone et al. (2019) from ObseRvations of Aerosols above CLouds and their intEractionS (ORACLES) and previous field campaigns. The impact of the discrepancy in SSA on the heating rate profile due to aerosols will be further discussed in Section 5.1.

          Only AOD is assimilated in MERRA-2 and therefore GOCART is used to distribute the aerosol within the

atmospheric column. The vertical profile of the mixing ratio of black and organic carbon in MERRA-2 is shown in Figure 2, alongside contours of cloud fraction from MERRA-2 with a value of 0.25. From an initial glance, it can be seen that larger values for the mixing ratio of black and organic carbon correspond to the dates with elevated AOD in Figure 1. The majority of the aerosol loading is located between 850 and 650 hPa, which corresponds to roughly 1500 to 3750 km in height in MERRA-2. In agreement with Figure 4 of Zuidema et al. (2018b), the black and organic carbon in MERRA-2 is located above

the cloud layer, but perhaps extends higher in the atmosphere than indicated by micropulse lidar observations (Figure 3). Qualitatively, MERRA-2 is also able to capture the thinning of the vertical extent of the aerosol as the loading decreases following the maximum in the middle of August.

The AOD at Ascension Island is a function of both the large-scale transport and also the timing and location of fires in Southern Africa. Some similarities can be seen between the back trajectories and the magnitude of the AOD at Ascension Island (Figure 4). The highest values of AOD were observed during August 2016 and September 2017. Both of these months have back trajectories that extend well into the African continent (Figures 4a and d), which is hardly the case for August 2017 when the subtropical highs over the southern Indian and Atlantic Oceans were shifted further to the east and the winds were weaker compared to 2016 (Figure 4c). Excluding the day with the highest AOD in August 2016, days with an elevated AOD

had a back trajectory that travelled from the south of Ascension Island, crossing the land-ocean boundary of the African coast between 10 and 15 °S (Figure 4a). On the contrary, days in August 2016 that observed an AOD below 0.3 tended to have back trajectories that originate further north.  The variance in daily AOD was not as large in September 2017, with most of the back trajectories having a more easterly path.

Given that the observed aerosol loading over Ascension Island is highest during August 2016, we have elected to

focus on that month. However, the same analysis has been completed for August 2017 as well as September 2016 and 2017 and monthly mean maximum SW heating rates within the atmospheric column due to clouds and aerosols for all months are presented in Table 3.

**4 Thermodynamic Profiles over Ascension Island**

A key characteristic of the atmosphere over Ascension Island is an inversion-topped marine boundary layer (MBL)

as seen in the August 2016 average temperature profile in Figure 5a.  Beneath the thermal inversion relative humidity is generally much higher and more hospitable for cloud development (Figure 5b). These features are present in both the INTERPSONDE observations and MERRA-2. However, there are differences between the two profiles. Within the boundary layer, MERRA-2 has a larger relative humidity, partially stemming from being slightly cooler than the observations. Perhaps due to the limited vertical resolution (there only eight model levels within the boundary layer), the inversion at the top of the

boundary layer is weaker in MERRA-2 and MERRA-2 is unable to capture decoupling within the boundary layer. More moisture is present in the middle troposphere, between 600 and 800 hPa in the observations. Finally, MERRA-2 has enhanced relative humidity aloft at 200 hPa, signaling the presence of clouds that are not detected by the observations. Excessive upper tropospheric cloudiness is a known feature in MERRA-2 (Bosilovich et al., 2015; Collow and Miller, 2016). A closer look at the temporal variation in the temperature and humidity profiles can be found in Figure 6. There is remarkable agreement

between the observations and MERRA-2. More interesting to note, is a connection between the relative humidity profiles and biomass burning aerosol overhead. As pulses of moist air become present in the middle troposphere with the entrance of a different airmass, so does the aerosol plume (Figures 2 and 6).

A thorough treatment of the thermodynamic structure during the biomass burning seasons of 2016 and 2017 can be found in Zhang and Zuidema (2019). MBLs of this depth typically accommodate transition cloud structure, which is characterized by single layer stratocumulus when the MBL is relatively shallow and trade cumulus when it is deeper. Intermediate stages in this deepening-warming MBL structure are characterized by hybrid cloud configurations consisting of a mix of layered stratocumulus and cumulus that intermingle in complex ways. Deeper MBL's tend to contain two or more internal boundary layers that are separated by a weak inversion, a process known as decoupling, which leads to the development of cumulus convection that rejoins the two-layers leading to a "cumulus-coupled" MBL. Manifestations of decoupling are best observed in the bottom panel of Figure 6, which exhibits a subtle, intermittent sub-layer at ~900 hPa, and in Figure 13 of Zhang and Zuidema (2019). Above the MBL where most of the biomass-burning aerosol is located, there are intermittent bursts of moist air, potentially a result of weak easterly waves. Occasionally these waves may be accompanied by mid-level cloud cover, for example at ~600 hPa around August 25, 2016, but these clouds are too thin and contain small enough droplets that they are not detectable using a cloud radar (see Figure 7a).

Cloud liquid water contents above the AMF1 from MICROBASE (Figure 7a) are derived by scaling the observed MWR liquid water path using a weighting function based upon the cloud radar effective reflectivity factor and an adiabatic assumption that utilizes constant cloud droplet number density. Thus, the assumptions in MICROBASE are consistent with adiabatic cloud liquid water being the dominant contributor to the retrieved effective radius relative to number density in the SW radiative calculations that follow. Figure 7a indicates cloud morphology that includes precipitating cumuli that are occasionally laterally detraining into an elevated layer of stratocumulus (August 25, for example). Cloud droplet effective radii are generally in the 5-10 µm range, although deeper plumes, such as those observed on August 28-29, exhibit elevated liquid water contents and cloud droplet effective radii that reach ~10 µm near cloud top (Figure 7b).

The clouds above the AMF1 site are primarily maritime as indicated by the occurrence of cloud base at the ocean lifting condensation level (not shown), but there is likely orographic enhancement from the island. Vertical velocities in the lower 600 m of the boundary layer above the AMF, as indicated by Doppler Lidar measurements, average 0.5 ms$^{-1}$ because of the continuous lifting imposed by the steep island orography immediately upstream. This lifting inevitably leads to modifications to the cloud structure. Most likely, the orographically forced updrafts enhance cloud development by lifting parcels from the ocean surface more readily to their LCL and reducing the rate at which precipitation reaches the surface by opposing the fall velocity of raindrops. The latter is confirmed by a zeroing in the mean sub-cloud Doppler velocity profile of raindrops above the AMF1 site at approximately 600 m (not shown). The almost certain increase in the fractional cloud cover relative to that in the undisturbed MBL implies that the heating rates in the presence of clouds presented in Section 5 are likely exaggerated relative to heating rates derived from radiative transfer calculations based on cloudiness over the ocean. Thus, the effect of clouds on the calculated heating rates above the AMF1 at ASI should be interpreted as an upper bound.

## 5 Results

## 5.1 SW Heating Rate Profiles over Ascension Island

Idealized radiative transfer calculations were used to quantify the heating rates of aerosols and clouds within the atmospheric column over Ascension Island. Given the discrepancy in SSA between MERRA-2 and the observations presented by Zuidema et al. (2018b), a sensitivity test was performed to determine the role of SSA on radiative heating due to aerosols within the column to quantify the uncertainty associated with the SSA used. Three different values of the SSA were used to represent the original SSA in MERRA-2, and potential deficiencies related to the vertical profile in relative humidity and the lack of brown carbon. In order to adjust for relative humidity, the SSA was determined by the lookup table used in MERRA-2 for the scattering and extinction properties of black and organic carbon at 550 nm as a function of the observed relative humidity. Adjusting for the humidity alone does not fully explain the difference in SSA between MERRA-2 and the observations, indicating that proper aging of the aerosol within the model is a necessity. To account for the lack of brown carbon, the SSA for organic carbon was multiplied by 0.85, which is the mean percent difference between MERRA-2 and the observations presented by Zuidema et al. (2018b). A summary of the monthly mean maximum SW radiative heating and where it occurred within the column for the entire 2016 and 2017 biomass burning seasons can be seen in Table 3 however the figures with more detailed information are only shown for August 2016 as that was the month with the highest aerosol loading.

Results for the SW aerosol radiative effect using these three sets of values for SSA under clear-sky conditions can be seen in the left column of Figure 8. Within the atmospheric column, the majority of the heating due to aerosols occurs in the layer around 800 hPa, though the impact of aerosols can be felt to a lesser extent aloft and down to the surface regardless of the SSA (Figures 8 and 9). There is minimal heating due to aerosols during the first few days of August 2016 as the AOD is only around 0.1. Throughout the rest of the month the radiative heating rate profile follows the periodicity of aerosol loading as seen in Figure 2 and the grey contours indicating AOD at 1 µm within the atmosphere in Figure 8. The contours for AOD are shown as a guide for the location of the aerosol. No conditional sampling for AOD was used for the calculation of the heating rates. Aerosols are spread within a deeper layer beginning 25 August 2016 and, as such, the heating within the column occurs in a thicker layer than earlier in the month. Although the highest AOD occurs on 13 August 2016, the maximum heating rate on that day is just shy of the largest heating rate of ~2.7 K day$^{-1}$ (~6.25 K day$^{-1}$ when the SSA for black and organic carbon is reduced) within the month that occurs on 30 August 2016 and 31 August 2016. A likely explanation for this is that there is a deeper layer containing aerosol at the end of August 2016.

As expected, heating rates are smaller when the original SSA from MERRA-2 is used. Though somewhat difficult to see with the color bar in Figure 8, but notable in Table 3, SW heating rates are slightly larger in magnitude when the SSA is scaled based on the observed relative humidity (Figure 8 a and c). The monthly mean maximum heating within the column due to aerosols is roughly a tenth larger with the relative humidity scaled SSA (Table 3). SW heating rates can actually double or triple if the SSA for organic carbon is reduced to simulate the role of brown carbon and to be more in line with the observed SSA (Figure 8b). This finding furthers the importance of an accurate representation of aerosol optical properties in models within the Southeast Atlantic already stressed in the literature (Mallet et al., 2019; Pistone et al., 2019; Shinozuka et al. 2020).

By comparing the results for cloudy to clear conditions, it can be seen that in the presence of clouds, radiative heating within the aerosol layer is embellished (Figures 8d, e, and f). This will be further elaborated upon later.

For simplicity, from this point forward heating rates due to clouds and aerosols are discussed using the relative humidity scaled SSA for organic and black carbon to present the middle of the road scenario that is observationally constrained along the vertical profile. Unlike in other regions, heating due to clouds, generally located around 900 hPa, is underwhelming (Table 3; Figure 9). There is less day-to-day variability in the magnitude of SW heating within the cloud layer compared to heating from aerosols (not shown). This is somewhat expected due to the consistent nature of the cloud water path and effective radius (Figure 7). There is however some variability in the location of the heating in connection with fluctuations in the height of the boundary layer, which could dampen out the local heating rates in a monthly average.

To isolate the absorption due to black carbon itself, the percentage of heating solely due to black carbon is shown as a percentage of the heating due to all aerosol species in Figure 10. Within the aerosol plume, between 65 and 80% of the SW heating is indeed a result of black carbon under clear skies (Figure 10). However, at the base of the aerosol layer, black carbon would actually produce more heating on its own had other species not been present. On occasion, percentages on par with what is in the aerosol plume itself can extend down to the surface. The remaining SW heating within the aerosol plume and down to the surface is likely due to the extinction of radiation from organic carbon that is not scattered within the plume. On days without an elevated AOD, such as the first few days in August 2016, there is a noticeable lack of heating due to black carbon within the column, especially in the boundary layer. In the presence of clouds, the percentage of SW heating due to black carbon is similar in magnitude to the clear sky case (Figure 9b). Differences arise at the base of the aerosol plume and in the boundary layer, as clouds become the dominant source of SW heating.

In terms of heating rates due to black carbon, our results are quite similar to those presented by Gordon et al. (2018), who showed a mean SW heating of 1.9 K day$^{-1}$ due to biomass burning aerosol for the period of 6 August 2016 through 10 August 2016 over the Southeast Atlantic. For the same five-day period, we see a mean daytime SW heating due to black carbon of 1.86 K day$^{-1}$ within the layer between 760 and 840 hPa. Gordon et al. (2018) took a similar approach by turning aerosols and black carbon off in a model simulation but this was done using global and regional simulations with HadGEM. However, the authors stated their results might not be representative of the heating that actually occurred as the aerosols in their simulations were too low in altitude.

There is an interplay between clouds and aerosols when they are considered together as opposed to individually. Photons scattered by clouds reenter the aerosol layer and have an additional opportunity to be absorbed within the atmosphere as opposed to reaching the surface. The enhancement of heating within the aerosol layer due to clouds is displayed in Figure 11 and is on the order of tenths of a K per day. On most days with sufficient aerosol loading the enhancement is a few tenths of a K per day but when all aerosols are considered the majority of the enhancement is located within the aerosol layer (Figure 11a). Additional heating due to aerosols in the presence of clouds occurs below the aerosol layer and down to the surface. This is limited to the morning and evening hours when the sun angle is low when all aerosols are considered, likely due to scattering from the abundance of sea salt in the boundary layer. (Figure 11a). There is some indication of an enhancement in SW heating

during the daytime hours when only black carbon is considered on days with high aerosol loading and in which the black carbon gets mixed into the boundary layer, such as 14 August 2016 (Figures 2 and 11b). However, there is likely not enough black carbon in the boundary layer for more of an enhancement to occur. Within the aerosol layer itself, between 900 and 700 hPa, black carbon is mostly responsible for the additional heating. The amount of enhancement in SW heating within the aerosol layer due to clouds is variable depending on the location and thickness of the cloud as well the AOD. The greater the AOD and cloud water path, the greater the interaction between the two. The largest local heating rate within the month, under clear skies, occurs on 30 and 31 August 2016. However, this occurs on 13 August 2016 when clouds are considered. The aerosol heating rate is further enhanced due to the presence of clouds on 13 August when not only the AOD is higher, but the cloud water content is also higher compared to the end of the month.

## 5.2 Direct SW Radiative Effect at the Surface and Top of the Atmosphere

The direct impact of aerosols on SW radiation at the top of the atmosphere (TOA) and surface can also be quantified, as shown in Figure 12. In this case, the direct radiative effect (DRE) due to aerosols is presented as a radiative flux in units of W m$^{-2}$ as opposed to a heating rate. Aerosols produce a cooling at both the surface and TOA, with a larger cooling under clear conditions. At the TOA, this is due to the additional scattering of SW radiation by clouds that then leaves the atmosphere at the TOA. At the surface, this is because without clouds present, SW heating due to aerosols also warms the boundary layer (Figure 9). The smaller the AOD is, the smaller the cooling, and the smaller the difference between clear and cloudy conditions. This is evident the first couple days in August 2016, as well as 19-22 August 2016. The two periods with enhanced AOD, 13 August 2016 and 26-31 August 2016 have a daily mean DRE due to aerosols at the TOA of ~20 W m$^{-2}$ under clear sky, however when clouds are considered, it is difficult to distinguish these days from the rest of the month. Cooling at the surface due to aerosols is larger in magnitude than at the TOA and reaches ~-40 W m$^{-2}$ on 13 August 2016 and the last few days of the month with clear skies, and -30 to -35 W m$^{-2}$ with all sky conditions (Figure 12).

The values for the all sky DRE at the surface across the entire month of August 2016 is similar in magnitude to what was presented by Chang and Christopher (2017). On the contrary, we show a cooling at the TOA while previous studies such as Zhang et al. (2016) and Chang and Christopher (2017) show a warming. Chang and Christopher noted the influence of the aerosol optical properties on the DRE effect at the TOA for radiation simulations of aerosol above clouds and performed the radiative transfer calculations using a fixed cloud optical depth and effective radius that was much larger than what was commonly observed at Ascension Island. Zhang et al. (2016) noted uncertainty in their calculations associated with the observations of aerosol optical properties, however in the southeast Atlantic they show a positive DRE at the TOA that lessens in magnitude as you move north and west from the African continent. The cloud albedo below the aerosol layer plays an important role in determining the sign of the DRE at the TOA (de Graaf et al., 2020). While the daily means in Figure 12 are negative, there are individual hours in which there is a warming at the TOA. Most notably, this occurs on 13 August 2016 and 28 August 2016 when the cloud water path and effective radius is above average compared to the rest of the month as seen in

Figure 7. As marine stratocumulus transitions to trade cumulus, breaks within the clouds are going to result in a DRE at the TOA that is overwhelmingly negative with sufficient aerosol loading. This means that at some point in the aerosol plume's progression westward across the Atlantic, the mean DRE at the TOA can switch signs.

## 5.3 LW Radiative Cooling over Ascension Island

Biomass burning aerosols tend to have a minimal direct impact in the longwave (LW) part of the spectrum, but they can indirectly impact the LW radiation within the atmospheric column. Heating within the atmospheric column can be lost due to additional LW radiative cooling in response to SW warming due to aerosols. At the present time, aerosols are not a direct input for RRTM LW. The observed temperature profiles were used as input to RRTM LW as a proxy for the presence of aerosols. In an effort to quantify the LW radiative cooling associated with SW aerosol heating, the hourly mean heating rates, as shown in Figure 8c, were subtracted from the observed temperature profile to represent a profile without aerosols; the humidity profile was not adjusted. The temperature profile was adjusted each hour, however, any SW heating that was not lost due to additional radiative cooling from the increased temperature in the run mimicking the inclusion of aerosols was allowed to persist through the following hour. The LW aerosol radiative effect is then considered to be the heating rate from the runs with the original temperature profile minus the heating rate from the run with the adjusted temperature profile. This methodology is somewhat extreme as heating due to aerosol can be transferred to other forms of energy such as latent heat and transported through advection. However, it can be used to determine whether the SW heating due to aerosols is offset by increased radiative cooling.

Results from this exercise, using the clear sky case, are displayed in Figure 13. Radiative cooling occurs throughout the aerosol layer and is maximized at the bottom of the layer, where at times it can locally reach near 3 K/day. Heat is transferred above and below the aerosol layer when the radiative cooling occurs, with a larger magnitude of the heat being displaced toward the surface. It is evident that without an atmospheric circulation or other processes occurring in the atmosphere, additional heat due to aerosol absorption remains in the column. This is demonstrated by the fact that radiative cooling still occurs through mid to late August despite a suppressed aerosol loading (Figure 13a). During the daytime hours, additional LW radiative cooling due to aerosols never offsets the absorption due to aerosols (Figure 13b). Even at night, the magnitude of the LW cooling due to aerosol never reaches the magnitude of the daytime SW aerosol heating. There is, however, a redistribution of heat as a result of aerosols. The largest magnitude of warming due to aerosol occurs during the daytime hours in the middle of the aerosol layer, and this daytime heating extends vertically in both directions. At night, cooling due to the SW absorption by aerosols is maximized at the bottom of the aerosol layer, though is present to some extent within the entire aerosol layer, and some heating occurs above and below the aerosol. The thermodynamic structure of the atmospheric column is therefore altered on a diurnal cycle when aerosols are present and this can have implications for other atmospheric processes such as the development, maintenance, and transition of marine stratocumulus and trade cumulus clouds (Zhang and Zuidema, 2019).

## 5.4 Radiative Heating Along a Back Trajectory

While it is informative to investigate the heating rate profile due to biomass burning aerosol above Ascension Island, it is imperative that such an analysis also be completed along the trajectory of the aerosol plume as it makes its way from southern Africa and over the Atlantic Ocean. A case study has been completed for the seven-day HYSPLIT back trajectory originating at 2 km above Ascension Island at 13z on 13 August 2016. This date was chosen as it had the highest observed and MERRA-2 analyzed AOD among the 2016 and 2017 biomass burning seasons. As indicated by the spread of the trajectories in panels a-d of Figure 5 from Zuidema et al. (2018b) and Figure 14, there is some uncertainty regarding the exact path of the biomass burning aerosol plume. In order to account for this, HYSPLIT was forced by the meteorology from the 27 ensemble members of NCEP's Global Data Assimilation System (GDAS) at 0.5-degree spatial resolution, in addition to MERRA-2. Clear-sky radiative heating rate profiles were then calculated along each latitude-longitude point of the back trajectories using the same methodology as for over Ascension Island except using the temperature and humidity profiles from MERRA-2. Shinozuka et al. (2019) demonstrated good agreement in the SSA between GEOS and aircraft observations over the Southeast Atlantic Ocean, unlike the discrepancy over Ascension Island, so the original MERRA-2 SSA was used. This is likely due to deficiencies in the MERRA-2 aerosol optical properties related to the aging of the biomass burning aerosol. Only clear sky was evaluated as MERRA-2 does not provide the necessary cloud microphysical parameters for RRTM. While there is the potential to gain this information from satellite observations, these observations would lack an appropriate vertical resolution and there would likely be inconsistencies between the thermodynamic profiles in MERRA-2 and the cloud structure in the observations. Given the RRTM results over Ascension Island, SW heating rates due to aerosols along the back trajectories are likely larger than what is presented for the clear sky scenario.

SW heating rates due to aerosols along the back trajectory can be found in Figure 15 a and b, respectively, for MERRA-2 and the GDAS ensemble mean. Given that the ensemble mean is shown for GDAS, the SW heating is overall smoother than for MERRA-2, however there is good agreement in both the magnitude and location of the SW heating. In expected agreement with Figure 8a, the maximum SW heating due to aerosols within the column at the onset of the back trajectory occurs just below 800 hPa with a magnitude of roughly 2.5 K per day. This heating spreads to a larger vertical area, in both directions, and increases to its maximum within the back trajectory by two days prior to reaching Ascension Island using GDAS and three days prior using MERRA-2. Heating is then minimized around four days prior to reaching Ascension Island. It is at this point there is considerable uncertainty in the back trajectories. As seen in Figure 14, there are some ensemble members from GDAS that loop to the north, a feature that is also present in the MERRA-2 back trajectory. A mismatch between the aerosol assimilation and the dynamics of the analyzed meteorology in both MERRA-2 and GDAS is evident by the decrease in AOD. This highlights the necessity of looking at SW heating along an ensemble of back trajectories.

As expected, based on the trajectories, there is minimal spread in the SW heating due to aerosols within the first few days before arriving at Ascension Island across the GDAS ensemble members. As time prior to the aerosol plume reaching

Ascension Island increases, so does the standard deviation of the SW heating due to aerosols. At four days prior to reaching Ascension Island, there is a noticeable increase in the standard deviation from the previous day, in coordination with the increased spread in the back trajectories themselves. Five days out there is a dipole in the height of the maximum standard deviation, with the standard deviation reaching 0.965 K day$^{-1}$ at 850 hPa. While a signature such as this is not noticeable in the

ensemble mean heating rate, it is present in the heating rate using the MERRA-2 back trajectory. This could perhaps indicate that the heating aerosol at 700 hPa is only present in the ensemble members that loop to the north, either as a result of the thermodynamic profile or the location of the aerosol. Greater than five days out, the location of the back trajectories are so varied that the standard deviation, nearing 1 K day$^{-1}$, is on par with the magnitude of the ensemble mean SW heating rate itself.

## 6 Summary and Conclusions

The interplay between clouds, aerosols, and radiation is a source of uncertainty within the atmospheric science community and within general circulation models, particularly in the southeast Atlantic region. In this study, an idealized approach was used to quantify the contribution of clouds and biomass burning aerosol to heating within the atmospheric column located above Ascension Island in connection with the LASIC campaign conducted by DOE's ARM program. The field campaign included the deployment of the AMF1 on Ascension Island that spanned two biomass burning seasons with the

highest aerosol loading present during August 2016 followed by September 2017. An additional focus was placed on determining the uncertainty in heating rates due to aerosols, whether related to the SSA or the trajectories used to represent path of the aerosol plume before reaching Ascension Island. An assessment of aerosols within the MERRA-2 reanalysis revealed good agreement in AOD compared to AMF1 and AERONET observations, likely due to the assimilation of AOD from MODIS. However, the SSA was too high in MERRA-2, impacting the absorption of SW radiation, and therefore heating,

within the atmospheric column. This was mitigated in the radiation transfer experiments by adjusting the SSA to be aligned with the observed relative humidity and reducing the SSA for organic carbon based upon observations to mock that of brown carbon. It is also possible that the vertical distribution of aerosol in MERRA-2 is not completely realistic as it does not contain an observational constraint.

Due to the uncertainty of the SSA, a range of possible SW heating rates due to aerosols were calculated. On average,

the maximum local aerosol SW heating within the column over the course of the biomass burning season likely ranges from 2 to 4 K per day. Local heating rates are sensitive to the thickness of the aerosol plume, as shown by Figure 8, and when integrated across the atmospheric column heating due to aerosols can be just as important on days that have a thick but not dense aerosol layer. There is variability in the heating due to aerosol as a result of day-to-day and seasonal fluctuations in aerosol loading and cloud cover as the large-scale circulation and presence of wildfires in southern Africa influence the AOD over Ascension

Island. Black carbon is responsible for up to 80% of the SW absorption within the aerosol layer, though clouds also contribute. Biomass burning aerosols and clouds are typically located in distinct layers during the months of August and September, though at times, biomass burning aerosols can extend to the surface. On days with the biomass burning aerosol plume overhead,

an enhancement of heating within the aerosol plume on the order of 0.5 K per day occurs with the presence of a cloud layer. Any heating within the atmospheric column due to aerosol is not offset by additional LW radiative cooling in response to aerosol SW absorption.

There are a few limitations to this study that are worth noting and perhaps expanding upon in future work.

1) Despite the fact that other processes within the atmosphere can respond to the presence of aerosol and the resulting heating, radiation transfer was isolated here. It was assumed that all SW heating would go into altering the temperature profile when in reality, some energy could be lost to other processes such as water phase transitions and anomalous ascent. Furthermore, vertical mixing was not accounted for, which can also alter the temperature profile.

2) The sensitivity experiments for SSA and the difference in the heating rate profiles when using the observed thermodynamic profile as opposed to MERRA-2 demonstrate how sensitive the heating due to biomass burning aerosol is to the optical properties of the aerosol. As the aerosol plume travels and ages, the optical properties become modified. This changes the scattering versus absorbing properties and indicates that the heating right off the coast of Africa can be very different compared to that over Ascension Island. In the trajectory simulations, the SSA may be appropriate off the coast of Africa, however as the aerosol ages, the optical properties used in the calculations become less and less appropriate.

3) A simplified representation of aerosols was used in the radiation transfer experiments. At the present time, RRTM only allows for one aerosol type to be characterized in each vertical layer. Not only did this result in a weighting of aerosol properties based on the species, but it also eliminated the ability to characterize the SSA based on wavelength. While a sensitivity study could be completed to quantify the impact of wavelength on the SSA, and therefore the heating rate, the results would likely not yield information that is any more realistic than what was presented here given the dominance of SW radiation centered around 550 nm.

One important implication of the present study is the dependence of the clear-sky (and presumably cloudy-sky) heating rates upon the exact trajectory experienced by the biomass burning plume as it moves from its source region to the Southeastern Atlantic. Clear-sky heating rates varied considerably depending upon trajectories dictated by the large-scale flow, which suggests that there may be an important scale interaction operating in this region. The length of the trajectory from the source region coupled with the loading of black carbon may be an important parameter in facilitating changes in the cloud structure across the Southeastern Atlantic. The most significant anthropogenic alterations to the natural stratocumulus and transition stratocumulus offshore might result from a plume that possesses a large amount of black carbon and follows a long trajectory across the stratocumulus region as it moves away from the African coast.

Ultimately, one goal of experiments such as LASIC and ORACLES is to determine how heating due to biomass burning aerosol influence the formation and transition of marine stratocumulus to trade cumulus. Toward that end, it is interesting to contemplate the potential implications of this study in that context. Immediately off the African coast in the region that experiences the strongest upwelling of cold ocean bottom water lies a shallow MBL containing predominantly single layer stratocumulus clouds. Absent warming above the plume by absorbing aerosol and associated increases in water vapor within the absorbing aerosol plume, these near-shore clouds are maintained by the production of turbulent kinetic energy

(TKE) by LW cooling at cloud top, which mixes the MBL, and they exhibit a strong diurnal cycle due to offsetting daytime SW warming near cloud top. Significant warming and moistening of the inversion above cloud top by an absorbing aerosol plume likely decreases TKE production at cloud top. This decrease is driven by a reduction of LW cooling at the cloud top due to warming and moistening the air mass above. Reductions in TKE by SW heating would likely result from absorbing

aerosols heating the air within the inversion and intercepting incoming solar radiation that would otherwise reach the cloud top. Past and recent modeling studies have shown that mixing at cloud top, alone, is unlikely to significantly alter the MBL cloud structure (Bretherton and Wyant, 1997; Kazemi-Rad and Miller, 2020). However, a reduction in TKE at cloud top could result in enhanced decoupling, which may alter the cloud structure. Reduced mixing in the MBL associated with decoupling may also reduce the surface latent and sensible heat fluxes, which are implicated in the modeling studies listed above as being

the key contributor to the transition of stratocumulus to cumulus. The transition from stratocumulus to cumulus over the Eastern North Atlantic in summertime is particularly sensitive to the Lagrangian derivative of the latent heat flux (Kazemi-Rad and Miller, 2020). Hence, applying similar logic to the study region, it is reasonable to postulate that elevated absorbing aerosols and any associated moisture plume following offshore trajectories and systematically reducing the surface latent heat flux along the trajectory would likely have the effect of delaying the transition to cumulus, which would be a cooling effect at the

ocean surface. The key connection in this hypothesis is the link between decoupling and ocean surface fluxes, which warrants additional investigation.

**Data Availability**

MERRA-2 data (GMAO, 2015a, 2015b) are available at https://disc.gsfc.nasa.gov/ and AMF1 data are available at

https://www.archive.arm.gov/discovery/.

**Author Contribution**

ABMC and MAM developed the methodology, completed the analysis, and wrote the manuscript. ABMC performed the RRTM runs and created all figures. LCT provided quality controlled INTERPOSONDE data and MPJ and MW processed and

provided data for the cloud fields. All authors except MW edited the manuscript.

**Competing Interests**

The authors declare that they have no conflict of interest.

**Acknowledgements**

This research was supported by the U.S. Department of Energy's Atmospheric System Research, an Office of Science Biological and Environmental Research program, under DE-SC0018274. Contributions from MJ and MW were supported by the U.S. Department of Energy's Atmospheric Radiation Measurement Facility and
Atmospheric System Research programs through Brookhaven Science Associates, LLC, under contract DE-SC0012704 with the U.S. DOE.

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

**Tables**

**Table 1: Shortwave radiation transfer experiments included in this study.**

| Experiment | Description |
|---|---|
| 1. Clean and Clear | Observed temperature and humidity profiles |

| 2. Smoky and Clear | 1. + All aerosol species from MERRA-2 |
|---|---|
| 3. Dirty and Clear | 1. + All aerosol species except black carbon |
| 4. Clean and Cloudy | 1. + Cloud observations |
| 5. Smoky and Cloudy | 2. + Cloud observations |
| 6. Dirty and Cloudy | 3. + Cloud observations |

**Table 2: Quantities calculated using the radiation transfer experiments and methods for their calculation using the numbered experiments in Table 1.**

| Calculated Quantity | Experiments Used |
|---|---|
| Clear Sky Aerosol Radiative Effect | 2 - 1 |
| Cloudy Sky Aerosol Radiative Effect | 5 - 4 |
| Clear Sky Black Carbon Radiative Effect | 2 - 3 |
| Clear Sky Black Carbon Radiative Effect | 6 - 5 |
| Enhancement of Aerosol Radiative Effect Due to Clouds | 5 - 2 |
| Enhancement of Black Carbon Radiative Effect Due to Clouds | 6 - 3 |

5    **Table 3: Monthly mean maximum heating rate within the column due to clouds and aerosols in K day$^{-1}$ and the pressure where the maximum occurs.**

| | August 2016 | August 2017 | September 2016 | September 2017 |
|---|---|---|---|---|
| **All Aerosols** | | | | |
| M2 SSA | 2.39, 840 hPa | 2.05, 870 hPa | 2.15, 870 hPa | 1.99, 663 hPa |
| M2 OC SSA * 0.85 | 3.41, 840 hPa | 2.40, 840 hPa | 2.43, 870 hPa | 2.32, 840 hPa |
| RH Scaled SSA | 2.48, 840 hPa | 2.05, 870 hPa | 2.12, 870 hPa | 1.99, 663 hPa |
| **Aerosols + Clouds** | | | | |
| M2 SSA | 2.64, 840 hPa | 2.29, 870 hPa | 2.44, 870 hPa | 2.29, 870 hPa |
| M2 OC SSA * 0.85 | 3.78, 840 hPa | 2.68, 840 hPa | 2.79, 870 hPa | 2.55, 840 hPa |
| RH Scaled SSA | 2.71, 840 hPa | 2.28, 870 hPa | 2.40, 870 hPa | 2.26, 870 hPa |
| **Clouds** | 2.32, 870 hPa | 2.20, 870 hPa | 2.20, 901 hPa | 2.13, 870 hPa |

**Figures**

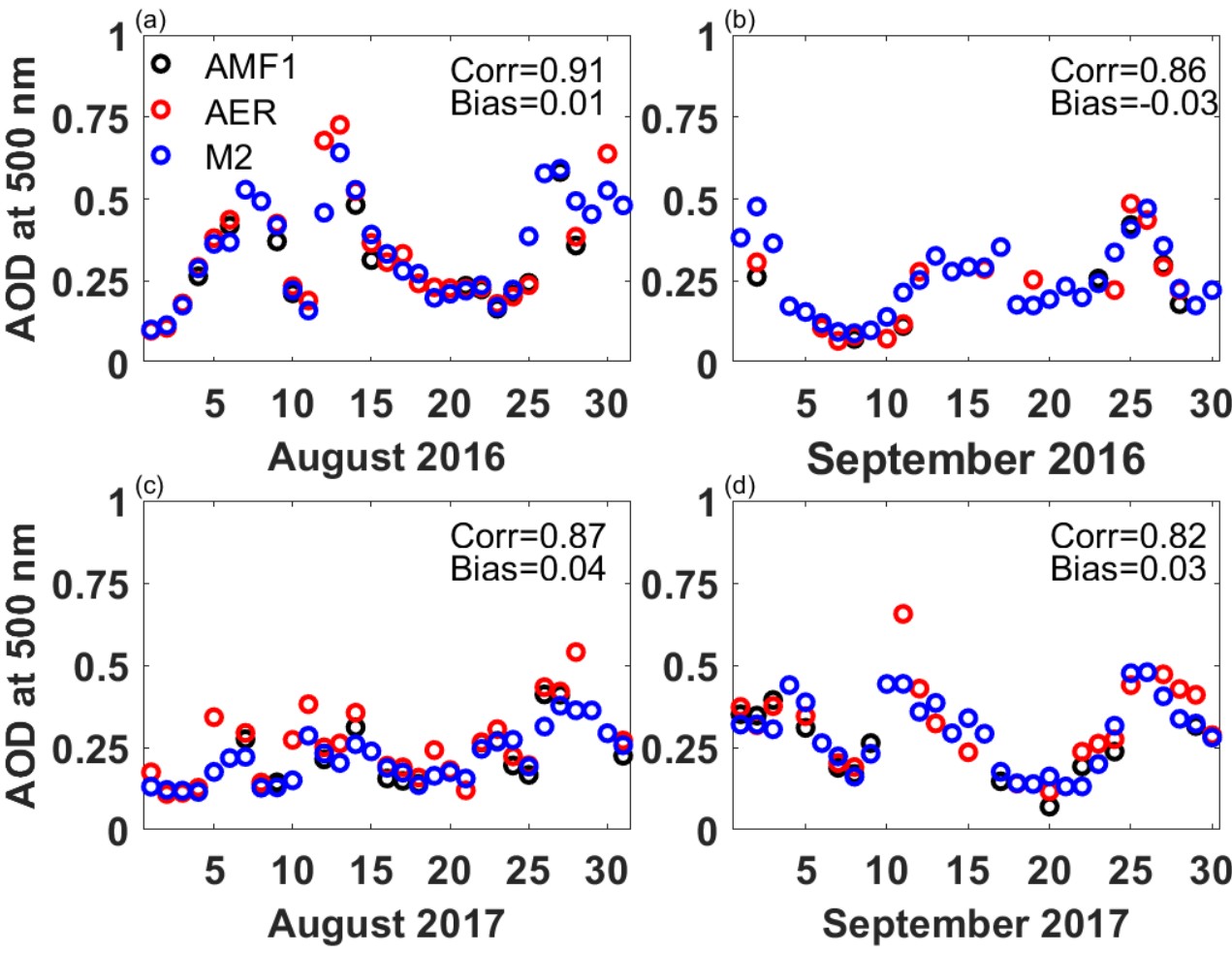

Figure 1: Daily mean aerosol optical depth from the AMF1 (black), AERONET (AER, red), and MERRA-2 (M2, blue) at Ascension Island during (a) August 2016, (b) September 2016, (c) August 2017, and (d) September 2017. Correlation and bias for MERRA-2 is with respect to AERONET observations.

**Figure 2: Vertical profile of the mixing ratio of black and organic carbon for the (a) 2016 and (b) 2017 biomass burning seasons from MERRA-2. Black contours indicate a cloud fraction of 0.25.**

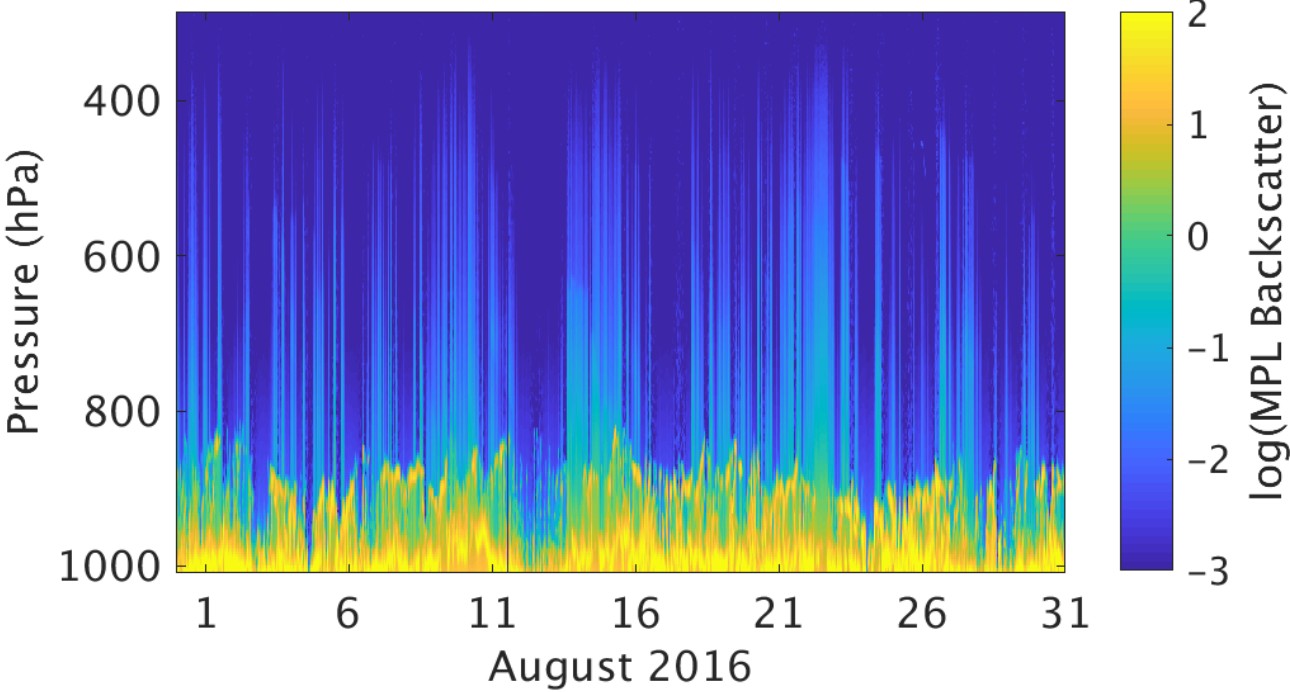

**Figure 3: The log of the micropulse lidar (MPL) backscatter over Ascension Island during August 2016.**

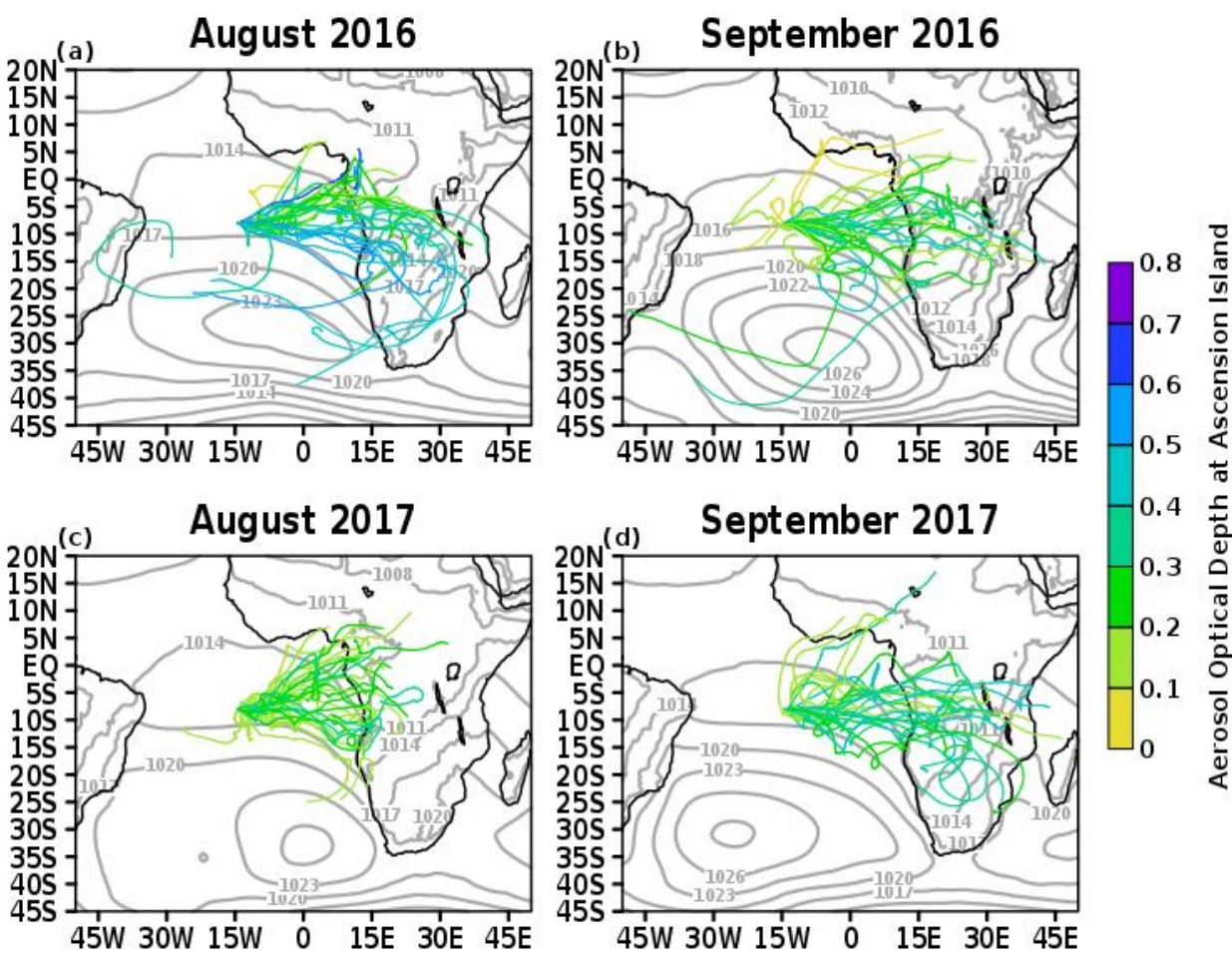

**Figure 4: 10-day back trajectories of a parcel originating at 2 km over Ascension Island colour coded based on the AOD on the start date for (a) August 2016, (b) September 2016, (c) August 2017, and (d) September 2017. Grey contours indicate monthly mean sea level pressure.**

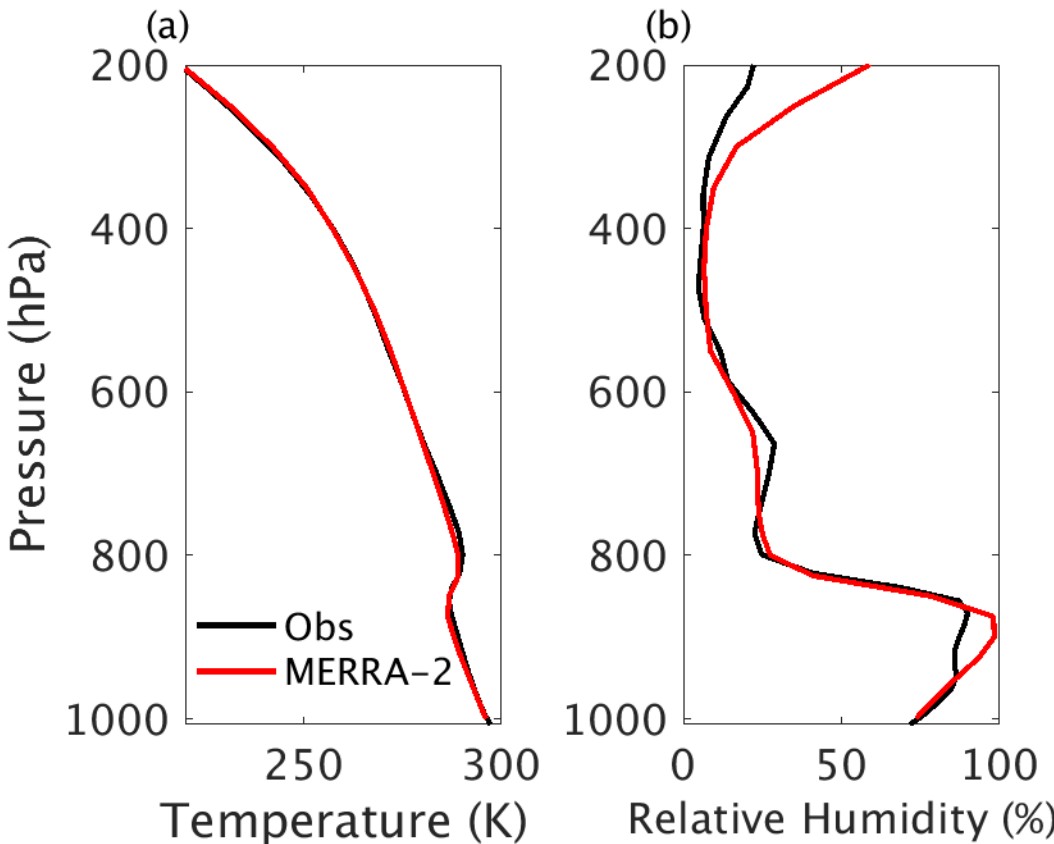

**Figure 5:** Average (a) temperature and (b) relative humidity profiles over Ascension Island from INTERPSONDE observations and MERRA-2 during the month of August 2016.

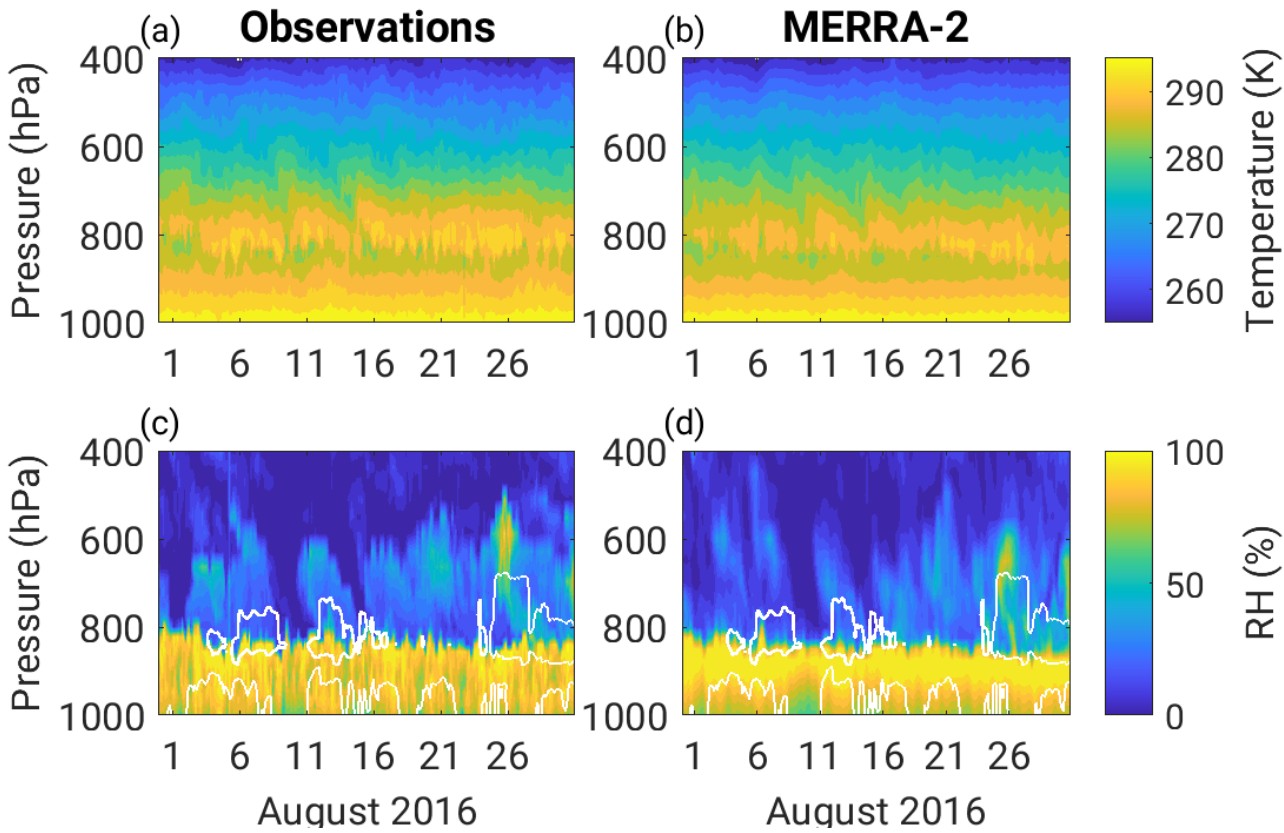

**Figure 6: Hourly vertical profiles of (a, b) temperature and (c, d) relative humidity over Ascension Island from the INTERPSONDE observations and MERRA-2 during August 2016. White contours in (c) and (d) indicate and AOD at 1 μm of 0.01.**

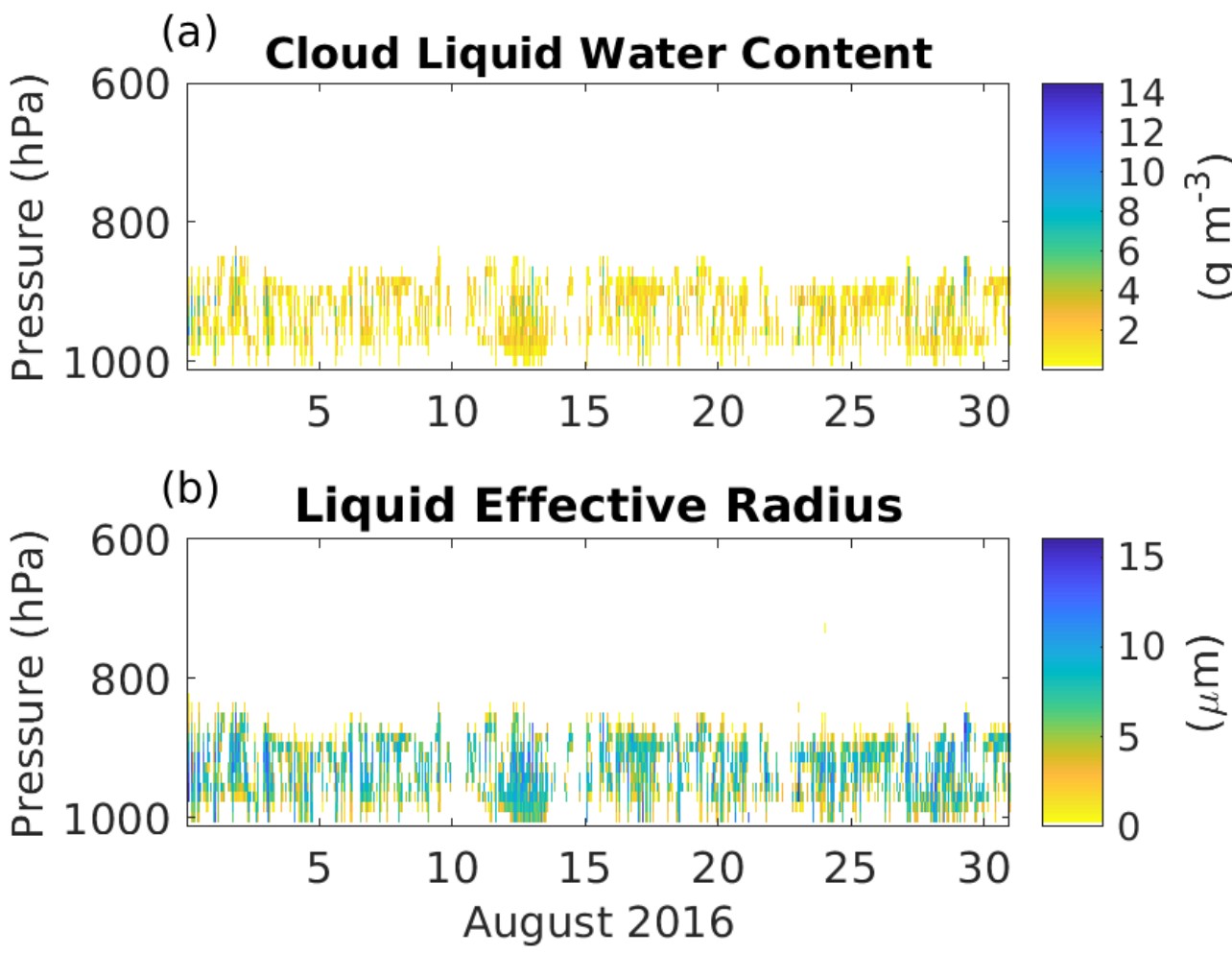

**Figure 7: (a) Cloud water content and (b) liquid effective radius over Ascension Island during August 2016 as calculated by the MICROBASE algorithm.**

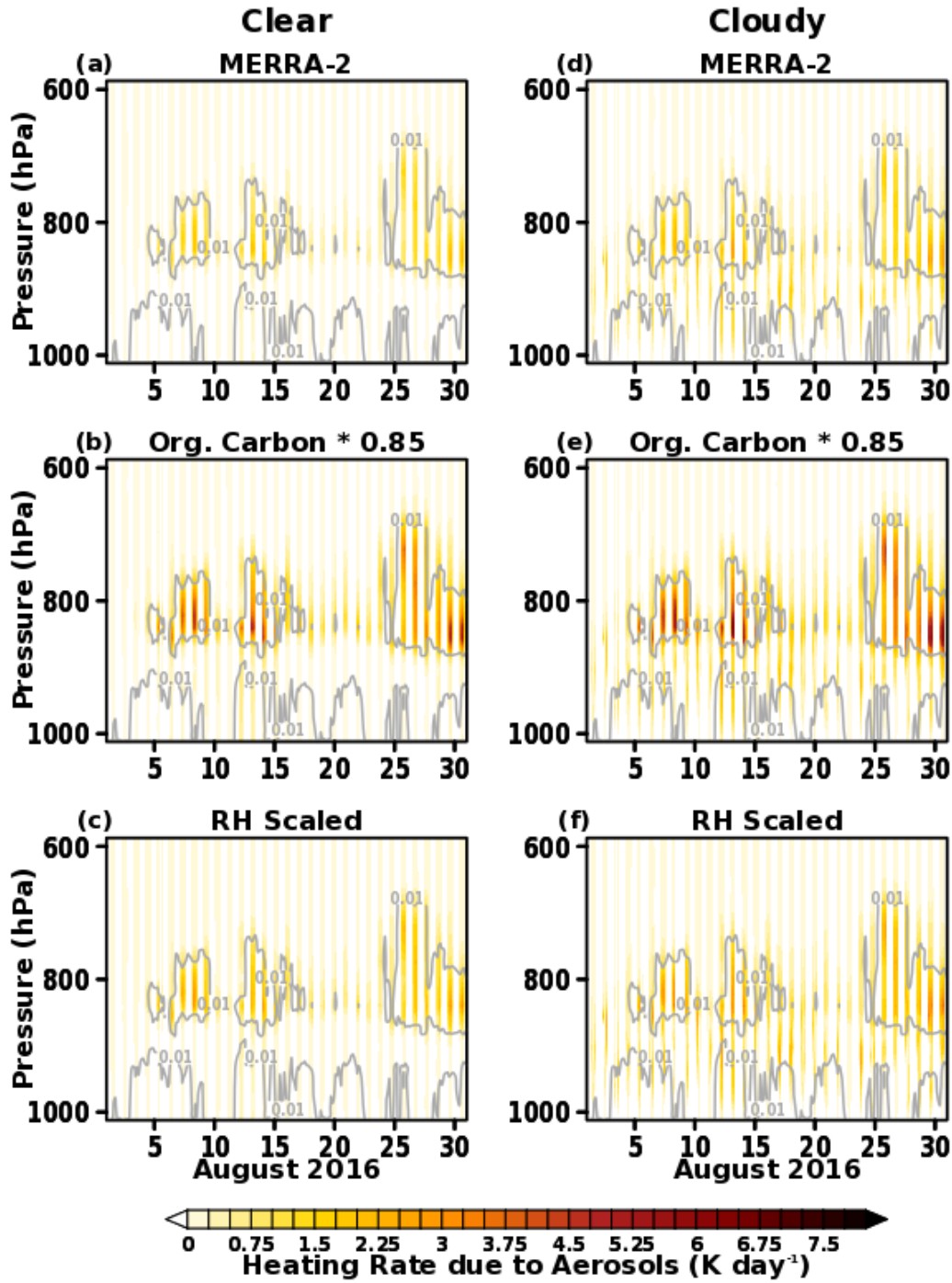

**Figure 8: SW heating due to aerosols based on the single scattering albedo (SSA) in (a, d) MERRA-2, (b, e) the SSA for organic carbon in MERRA-2 multiplied by 0.85, and the SSA in MERRA-2 rescaled based on the observed humidity profile over Ascension Island during August 2016 under (a, b, c) clear and (d, e, f) cloudy skies. Grey contours indicate an AOD at 1 μm of 0.01.**

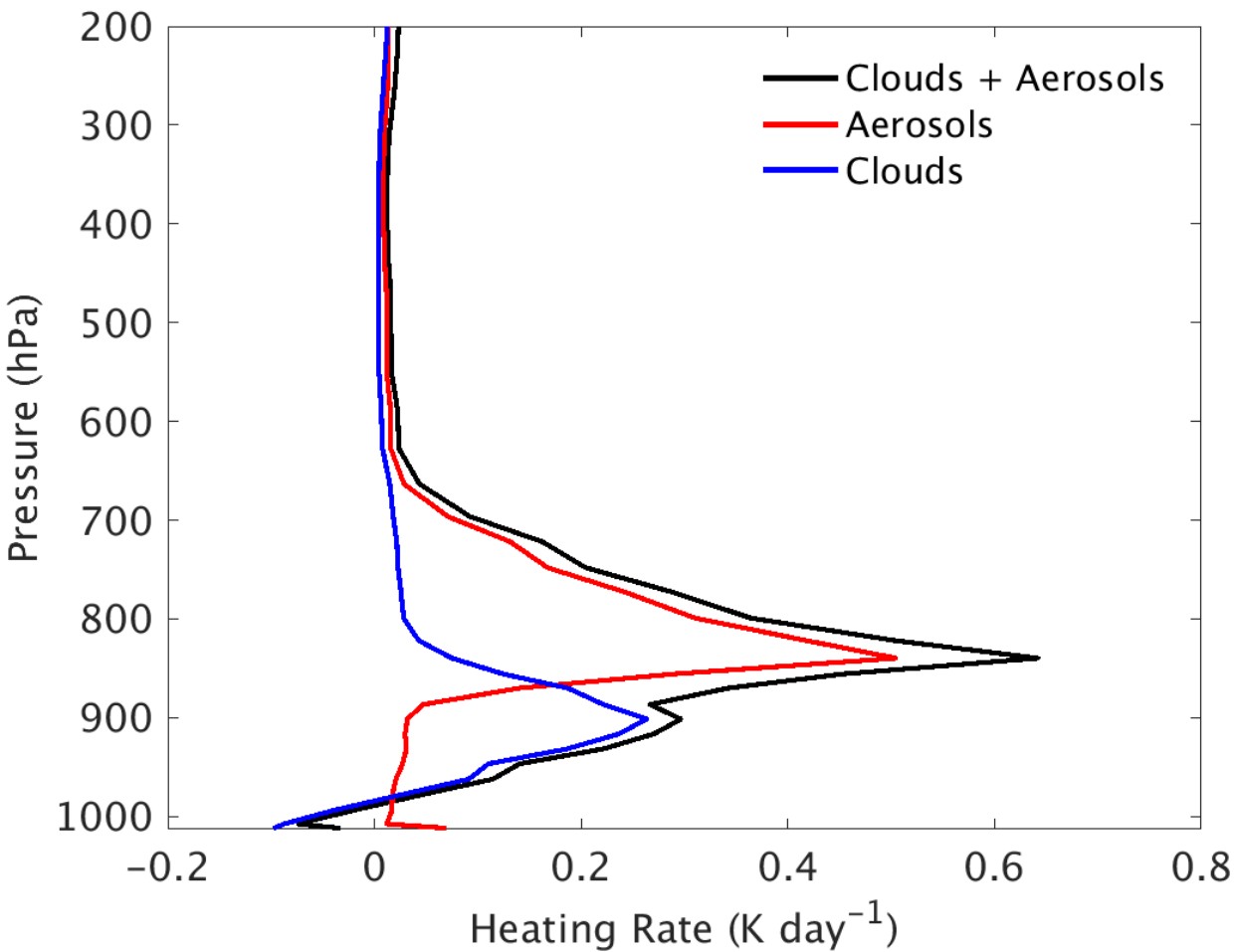

**Figure 9: Month averaged profiles of SW heating due to aerosols, clouds, and clouds plus aerosol during August 2016 over Ascension Island.**

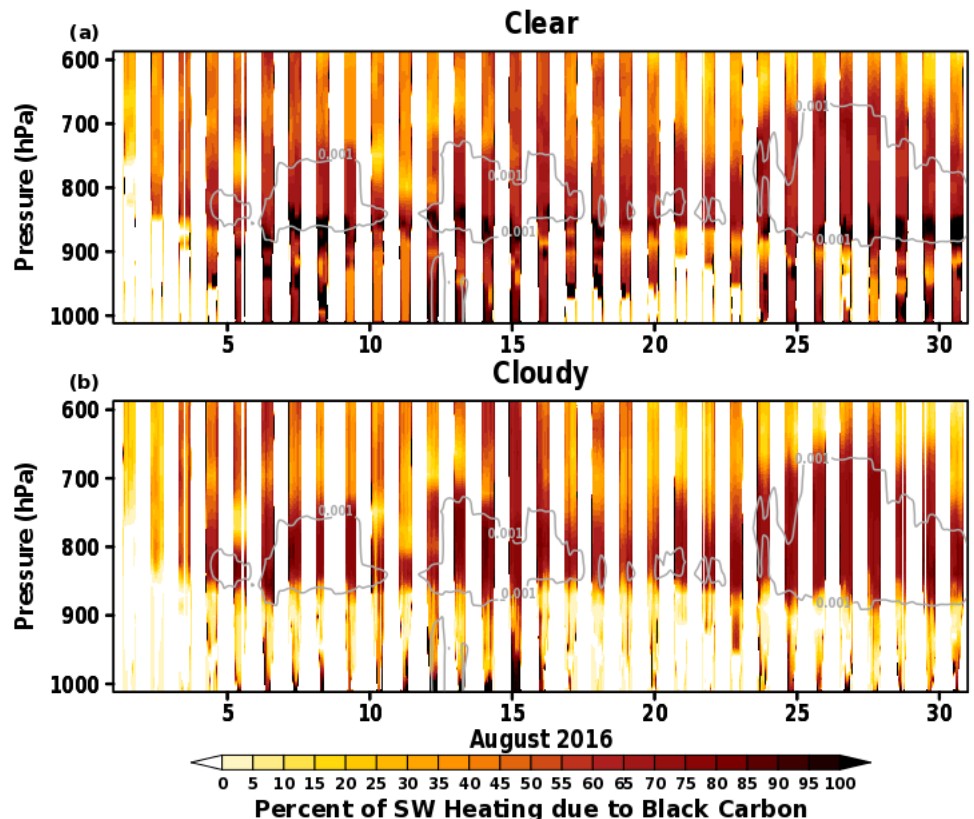

**Figure 10: Percent of total SW heating due to black carbon under (a) clear and (b) cloudy skies over Ascension Island during August 2016 using the SSA scaled by relative humidity. Grey contours indicate an AOD at 1 μm for black carbon of 0.001.**

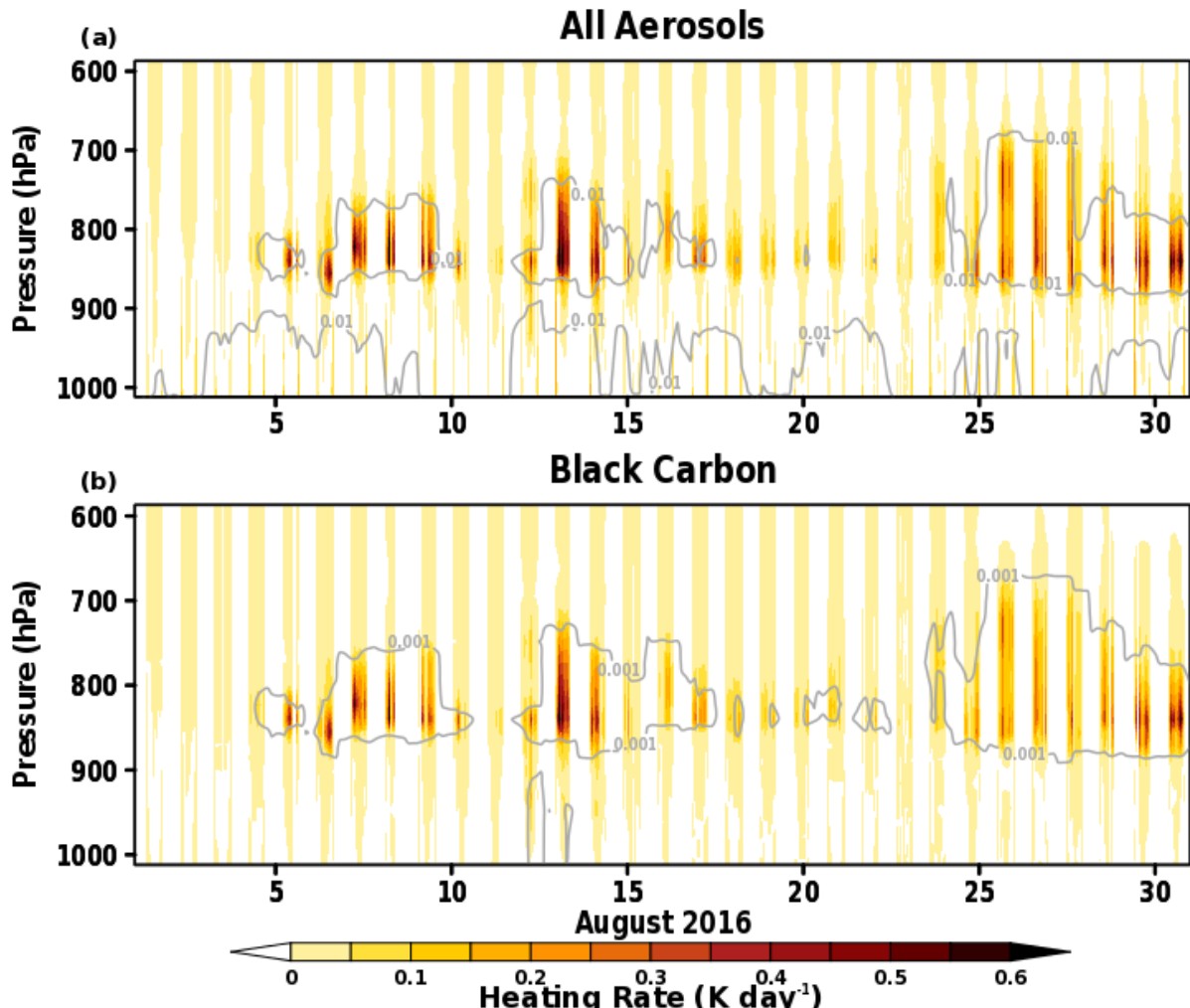

**Figure 11: Enhancement of SW heating due to (a) all aerosols and (b) black carbon in the presence of clouds over Ascension Island during August 2016 using the SSA scaled by relative humidity. Grey contours indicate an AOD at 1 μm of 0.01 in (a) and 0.001 in (b).**

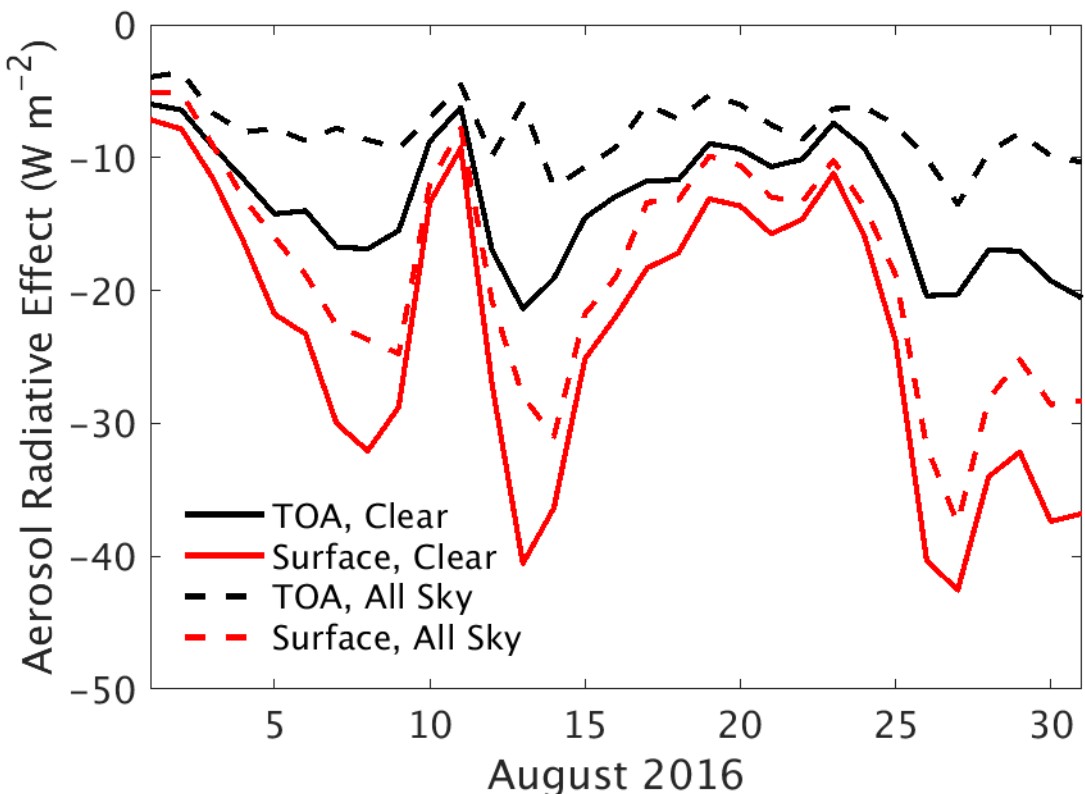

**Figure 12: Direct radiative effect due to aerosols at the surface and top of the atmosphere under clear and all sky conditions during August 2016 over Ascension Island.**

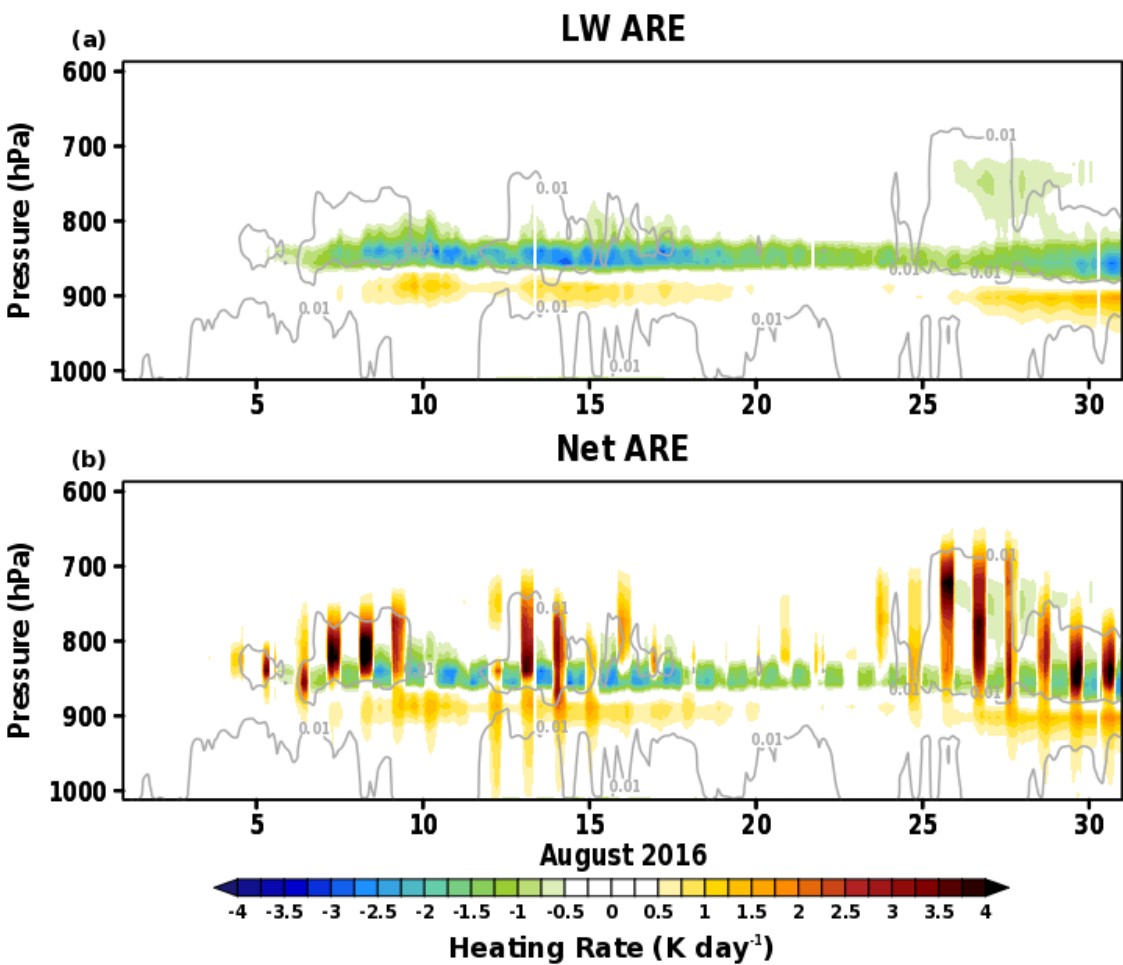

**Figure 13: (a) LW cooling as a result of increased temperature from SW heating due to aerosols with the relative humidity scaled SSA and (b) the net heating rate due to aerosols over Ascension Island during August 2016 under clear skies. Grey contours indicate an AOD at 1 μm of 0.01.**

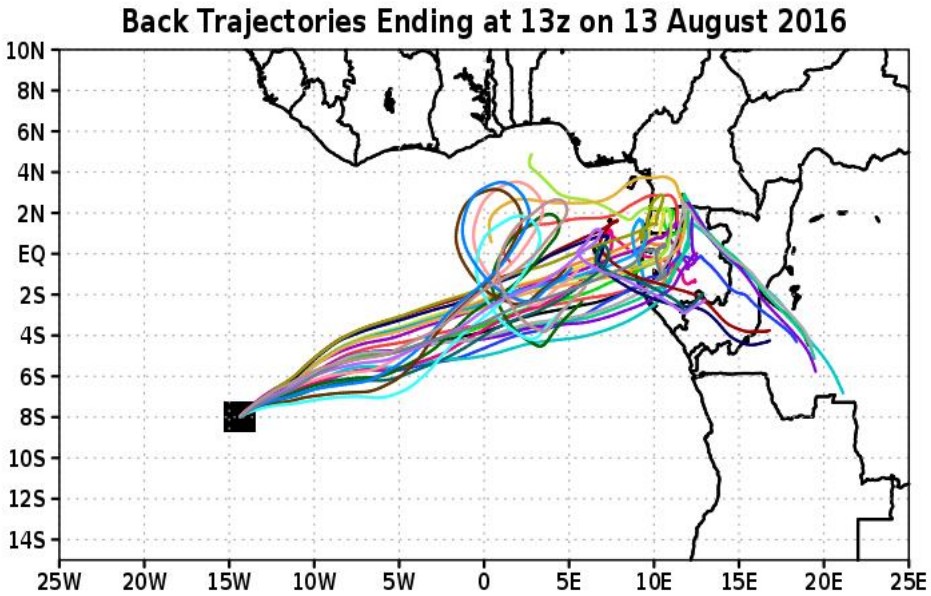

**Figure 14: Seven day HYSLPIT back trajectories forced with meteorology from the 27 ensembles of the GDAS, originating at 2 km above Ascension Island at 13z on 13 August 2016.**

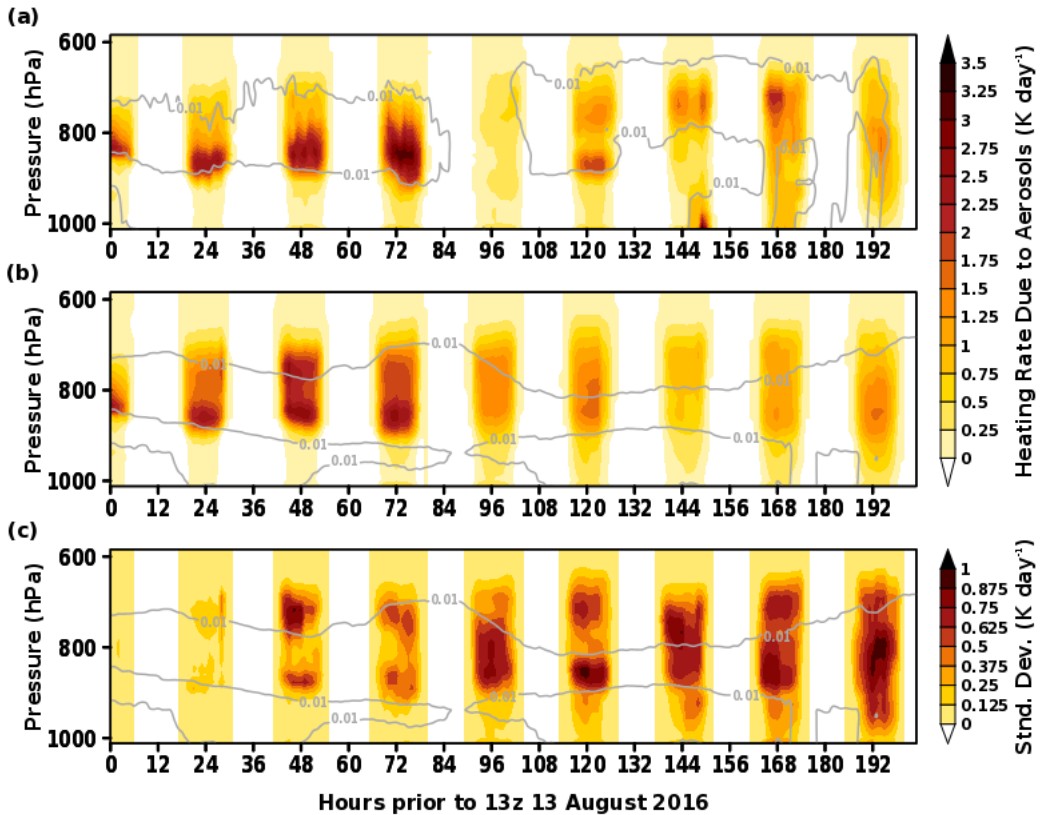

**Figure 15: The SW heating rate profile due to aerosols (a) along the MERRA-2 trajectory (b) the mean along the ensemble of back trajectories displayed in Figure 10 using the GDAS ensemble members and (c) the standard deviation originating at 2 km over Ascension Island at 13z on 13 August 2016.**

