# Peer review of "Radiative Heating Rate Profiles over the Southeast Atlantic Ocean during the 2016 and 2017 Biomass Burning Seasons"

_Atmospheric Chemistry and Physics, 2020_

## Referee Comment (RC1) · Anonymous Referee #1 · 13 Mar 2020

**Review of "Radiative Heating Rate Profiles over the Southeast Atlantic Ocean during the 2016 and 2017 Biomass Burning Seasons" by Collow et al.**

This manuscript presents a quantitative assessment of the radiative heating rates due to aerosols over Ascension Island based on idealized calculations through a Radiative Transfer Model, using aerosol vertical profiles and optical properties from MERRA-2 reanalysis, thermodynamic profiles, and low-cloud properties from island-based observations during the LASIC field campaign. The authors find shortwave heating within the aerosol layer above Ascension can locally range between 2 and 8 K per day, and shortwave heating due to biomass burning aerosols is not balanced by additional longwave cooling.

The presented assessment of the aerosol radiative heating rates is novel and nicely conducted, which could be informative and insightful to the scientific community if the following concerns and questions are properly addressed and justified. In addition, I find myself having hard times understanding/digesting many of the discussions in the current form of the manuscript. These confusing discussions should be reconstructed and extended with additional details and elaborations before this manuscript can be accepted for publication.

**Major concerns/questions:**

1.  The current introduction section seems a little weak in terms of scientific motivations. I recommend stating a stronger scientific motivation for the study, clarifying the scientific question you want to address with this study, and elaborating more on how is your study going to help us better understand the aerosol-radiation-cloud interactions within the SE Atlantic, rather than saying that the goal is to quantify and report the radiative heating rates over Ascension.
2.  You compared vertically integrated aerosol properties from MERRA-2 to LASIC measurements, but not thermodynamical properties, especially thermodynamic profiles, which could have subtle impact on the vertical distribution of biomass burning aerosols.
    a.  You mentioned in Line 20 on Page 11 that "boundary layer is too deep over Ascension Island in MERRA-2," comparing to LASIC measurements? Could you show the thermodynamic profiles comparison over Ascension?
    b.  You mentioned potential deficiencies in RH profiles of MERRA-2, can this also be shown as a comparison with LASIC measurements?
3.  You introduced six sets of experiments towards the end of Section 2. Please specify how are they defined/selected?
    a.  Are you using cloud observations to screen for clear skies?
    b.  Are you simply setting cloud parameter off in the model for clear sky case, even though the inputting thermodynamic profiles felt the existence of a cloud layer? If this is the case, please discuss how is this artifact affecting your results (i.e. is your heating rates under/over-estimating?).
    c.  Not all of these experiments are presented in the following results part, e.g. the clean-cloudy case is not shown.
4.  Section 3 is named as "Results," even though 3.1 and 3.2 are not actual "results" of the radiative transfer calculations. Using "Results" as a section name is vague, and I suggest reconsidering the section organization, one option could be making 3.1 as a new Section 4, and 3.2 as a new

Section 5. Currently, all your results are lumped into one section (3.3), which is poorly organized, I suggest making 3.3 as a new Section 6, and break each part into subsections, e.g. SSA sensitivity, SW heating due to black carbon, SW heating enhancement in the presence of clouds, LW cooling, and heating rates along back-trajectories.

5. I am a bit lost regarding the purpose of discussing the back-trajectories, how are they (or the origin of aerosol) related to (or affect?) your heating rate calculations? Besides, some of the details regarding the HYSPLIT runs should be introduced in the Data & Methodology section. P7 L15-18 seem abrupt at the end of Section 3.1, they also fit better in the Data & Methodology section. Perhaps these last two paragraphs of Section 3.1 could be moved to Section 2 together.
   a. P7 L2, "determine the origin of the aerosol" is a strong statement, as back-trajectories not necessarily indicate the exact pathways of biomass burning plumes, and one has to combine other sources, e.g. fire emission data, in order to determine the origin of the aerosol observed at Ascension. I suggest rewording.
   b. P7 L5-6, there were aerosol both in the free-troposphere and boundary layer, why 2 km is picked? What is the prior results you are referring to, please provide the reference.
   c. P7 L9-10, why the subtropical high over the southern Indian Ocean plays a role here? Monthly mean SLPs could be added to the background of Fig. 3 to base this statement.
   d. P7 L10-14, what are these observations implicating, how are they related to this work, or affecting the interpretation of your results?

6. Regarding Fig. 4, although one can identify an inversion layer from temperature profiles (distinguishing yellow and light orange from your plot), I would argue that potential temperature is a better choice to show the cloud top inversions. I think you could also extend the discussion to the potential role of the RH plot on indicating the biomass burning smoke plumes arriving at Ascension, as RH bursts in the free-troposphere tend to colocate well with the smoke episodes arriving at Ascension.

7. You did not discuss Figure 5b at all, and only mentioning Fig. 5a for clouds that are not visible in the plot. I wonder if this figure is necessary, it doesn't seem to add much useful information, and you barely discussed it in the main text.

8. The discussion in the first paragraph of Page 9 is very confusing to me. In the first couple of sentences, you mentioned comparing heating due to clouds with heating due to aerosols, and my interpretation of Figure 6d is that this is SW heating due to aerosol under a cloudy sky, and you did not show a case for heating due to clouds alone (no aerosol), so how did you make the comparison? whereas in the last sentence you talked about comparing aerosol heating under cloudy and clear skies (isn't this contradicting to the first couple of sentence? please clarify), and what do you mean by "embellished," I had trouble relating this word to the observations.

9. I also think the discussion for Figure 6d,e,f could be substantially extended. Currently, you barely discussed them ("embellished" is all you used to describe the comparison), and I will be curious to know why SW heating due to aerosol in the BL is enhanced under cloudy conditions? This is contradicting to my intuition, as the cloud layer reflects SW back to the space, I would expect the SW heating in the BL to decrease instead of increase.

10. It seems to me that MERRA-2 is not distributing enough BC/OC in the BL based on your Figure 2, 6 and 7 (clear sky condition). I wonder if you have compared MERRA-2 aerosol vertical profiles with extinction profiles from LASIC MPL (when available) or NASA ORACLES HSRL2 profiles or UK CLARIFY EXCALABAR profiles when they become available over Ascension during the 2017 season?

11. Regarding the LW cooling associated with the SW heating, the concern I have is that the observed temperature profiles (from LASIC radiosondes) had already felt that heating, in other words, the temperature increase was already taken into account in the observed profiles. By modifying the observed T profiles, you're artificially increasing the temperatures (artificially boosting the LW cooling). I would recommend just simply turn on the LW calculation using the same observed T profiles, and see if the net radiation budget produce a cooling or a warming.

12. You should state whether this LW experiment is calculated with cloud presence or not.

13. The discussion in lines 17-32 on page 10 is particularly hard for me to digest. As we know LW cooling is always happening no matter aerosol presents or not, and the "LW cooling" you are talking about in this paragraph is the additional LW cooling caused by the increase in the temperature profiles due to SW heating (since it is done by subtracting a control run). The following points should be addressed properly in order to make the discussion clear.

    a. When you say "LW radiational cooling never offsets the absorption due to aerosols," you should make this clear that you mean the additional LW cooling never offsets the absorption.

    b. L 25, "magnitude of the LW cooling never reaches …" same problem as above. LW cooling at inversions can easily reaches 10K/day at night. The LW cooling you are referring to is the difference between the T-modified run and the control run. Please make this clear.

    c. L 23, "radiational cooling still occurs…" As mentioned above, LW radiative cooling always occurs no matter the aerosol condition. Since you are showing the difference between the T-modified run and the control run, as long as there is SW heating due to aerosol (no matter how much), T profile will be modified, and difference in LW heating will exist. This cannot be used to demonstrate that additional heating due to aerosol remains in the column, you have to use the real LW heating profile to quantify that, not the difference between two runs.

I strongly recommend re-assessing this LW part (see Major comment 11), at least the way you interpret/discuss it.

14. Regarding your case study on the back trajectory, first, please specify reasons for originating at 2 km, second, why the meteorology for HYSPLIT runs are switched to GDAS instead of MERRA-2 as you did for the monthly back-trajectories, is MERRA-2 not capable to do ensemble runs? Please justify. Then, trajectories were forced by GDAS but radiative transfer calculations were using MERRA-2 thermodynamic profiles (why inconsistent)?

15. Why is the SW heating along the back trajectory limited to below the inversion? I would expect there to be aerosol in the FT along the 7-day back trajectories, and why not showing the aerosol and thermodynamic curtain plots along the trajectories from MERRA-2? In Lines 18-20, you're saying the aerosol layer is entirely above the inversion along the trajectory, and yet, no heating above the inversion? This is very confusing, please justify.

16. In the last paragraph, you mentioned that the ultimate goal is to study how the heating due to aerosols impacts the transition of marine stratocumulus to trade cumulus, I would really love to see more discussions added to the manuscript on how will this study help towards achieving this goal. For instance, how can this study contribute to the understanding of cloud adjustments to aerosols, and what insights can this study provide on the stratocumulus to cumulus transition in the southeast Atlantic. Such discussions will substantially strengthen the scientific importance of this study.

**Minor issues:**

*Abstract*
- Line 30, you mentioned "stabilization of the lower troposphere," but this is not discussed anywhere in the main text of manuscript. I suggest adding discussion regarding this point you raised in the abstract.

*Introduction*
- P2 L21, you haven't introduced Ascension Island yet, a general reader would have no idea where the island is, near coast? or in the remote ocean? I suggest introducing Ascension Island somewhere in the introduction.
- P3 L10-15, these information on datasets belong to the Data section, seems to me.
- P3 L16-18, these sentences seem to belong to the Methodology section. I would suggest adding more motivational statements, clarifying your scientific goals, here in the last paragraph of the introduction, replacing these details of datasets and approaches.

*Data and Methodology*
- P4 L7-, Because of the location of the AMF1 site, orographically generated clouds frequent present in LASIC AMF1 cloud measurements, please address how will this feature affect your assessment and the general representativeness of your results. Please also specify the temporal resolutions of ARSCL and MICROBASE.

*Results*
*3.1 Evaluation of Aerosols in MERRA-2*
- P6 L23, my understanding of the location of AMF1 site is that it is elevated and located at the upwind part of the island, which should be representative of the aerosol condition of a marine boundary layer (minimal island effect). Besides, if indeed there were dust (more scattering) mixed into the AMF1 sampling volume, shouldn't we expect a higher SSA? (Zuidema et al. 2018b's values are lower). Please correct me if this is not the case.
- P6 L28-29, could be helpful if AODs are overlaid on top of Fig. 2.
- P6 L34, the decrease in BL height is not very evident based on Fig. 2, overlaying some other forms of indication could be helpful.
*3.2 Thermodynamic Profiles over Ascension Island*
- P7 L21, "time-height" should be time-pressure, as you're showing pressure in the vertical.
- P7 L22, cloud top inversions at Ascension are not around 700 hPa (~3 km). Please double check the pressure axes in Figure 4.
- P7 L31, I do not see a "subtle, intermittent sub-layer at ~900 hPa" based on the RH curtain plot, perhaps this will be more visible in a single-profile presentation. Based on Fig. 13 of Zhang and Zuidema (2019), the intermittent layer seems to be located at ~700 m, which is lower than 900 hPa.
*3.3 Heating Rate Profiles over Ascension Island*
- P8 L21, it would be easier to visualize this co-variability between the heating rate and the AOD, if you could add MERRA-2 aerosol contours or AOD time series in the background of Fig. 6.
- P8 L29, if it is hard to tell with the color bar, could you provide some values to indicate the difference between heating rates calculated from MERRA-2 SSA and RH scaled SSA?

- P9 L18-19, I see heating due to black carbon ~0.5 K/day extending to around 600 hPa just as in Figure 6, please re-state your argument about this observation. Besides, how do you know it is absorption from dust (isn't dust more scattering)? Please justify.
- P9 L29, please define "enhancement of heating within the aerosol layer due to clouds" in the text or in the caption, i.e. how did you quantify that, is this cloudy-aerosol run minus clear-aerosol run (Fig. 6f – Fig. 6c = Fig. 8a?)?
- P9 L30-32, "a few K per day"? the color bar on Fig. 8 only goes to 0.6, how did you get a few K per day? "…but when all aerosols are considered the majority of the enhancement is located …" isn't this true for both 'All Aerosols' and 'Black Carbon'? Please check your logic here.
- P9 L32-34, could you please extend this discussion, especially on why this BL enhancement is not apparent for Black Carbon only case, even though BC is highly absorptive?
- P10 L1-2, I think you should be careful here and say "…due to the presence of clouds…"
- P10 L9 and thereafter, you used the phrase "radiational cooling," while I am more used to seeing "radiative cooling" being used in other literatures.
- P10 L28-29, "some heating occurs above and below the aerosol," we can't tell where the aerosol layer is based on this plot, one option is to put MERRA-2 aerosol contours in the background. Another option is to show a line plot highlighting a single heating profile along with the aerosol profile.
- P10 L29, please discuss how is this redistribution of heat, as you put it, going to modify the stability of the boundary layer, as you pointed out in the abstract.
- P11 L21, please elaborate more on how the depth of BL affects the SW heating. Why minimal SW heating occurs in the last few hours of the back trajectory?
- P11 L23, please explain or discuss your speculations on why SW heating maximized at the surface here.

*Summary and Conclusions*
- P12 L12, "…greater depth of the boundary layer…" comparing to what?
- P12 L16, "local heating rates are sensitive to the thickness of the aerosol plume" this is not discussed in the results section. Which figure supports this argument? Please mention this argument when you discuss that figure.
- P12 L20, "…most of the SW absorption" please be quantitative here.
- P12 L21, please be specific about this statement, i.e. which month? over Ascension or the whole SE Atlantic? Zuidema et al. 2018b states smoke often presents in the BL of Ascension Island, more frequent than "at times."
- P12 L23-24, I think what you want to express here is that adding a cloud layer will result in an enhancement of heating. Saying "interaction between SW radiation, clouds, and aerosols" is a bit misleading, as aerosols and clouds are not interacting in your calculations. I suggest a more careful rewording.
- P12 L29, I didn't think you were trying to represent the entire southeast Atlantic using Ascension observations until I saw this statement, and I don't think this study should be used to represent the entire region. I suggest stating this clearly in the introduction or data section, that this study only represents the remote SE Atlantic, and cannot be used to represent the entire region. Also, you could change the title to "…remote SE Atlantic…"
- P12 L31, "sensitive the heating … is," sensitive to what?

- P13 L15-20, the first and third sentences are the same sentence, please double check.

**Figure/Table issues**
1. Table 1, there are no italicized values in this table, please check.
2. Figure 2, please use $10_{-5}$ instead of e-05
3. Figure 4, please double check pressure axis, you are showing a 3 km BL.
4. Figure 5, in my opinion, this figure can be removed, and all the white space above 800 hPa can be minimized.
5. Figure 6, SSA instead of "SSA albedo." Again, the space above 600 hPa can be minimized, same for Fig. 7, 8 and 9.
6. I think for Figure 7 and after, you probably should remind the reader that we should compare these results only with the bottom panel of Fig. 6 (the RH scaled one), by making a note in the caption that SSA is the RH scaled one.
7. Most of the results are presented in curtain plots, they are nice in terms of showing the whole month, but rather poorly representing details in vertical. I recommend showing couple plots with single profiles when you discuss details in vertical, especially when you discuss the relative location of heating/cooling to the aerosol layer.

---

## Referee Comment (RC2) · Anonymous Referee #2 · 17 Mar 2020

Review on "Radiative Heating Rate Profiles over the Southeast Atlantic Ocean during the 2016 and 2017 Biomass Burning Seasons" by Collow et al.

This study attempts to quantify the contribution of aerosols and clouds on radiative heating rates within the atmospheric column over Ascension Island in South-east (SE) Atlantic. The approach involves the use of thermodynamic profiles and low-cloud observations during LASIC field campaign and aerosol profiles from MERRA-2 reanalysis data as inputs to a Radiative Transfer Model (RRTM). The study finds that on average, the maximum local aerosol SW heating within the column over the course of the biomass burning season ranges from 2 to 4 K per day. In addition, on days biomass burning aerosol plumes are observed above clouds, shortwave heating within the aerosol plume is enhanced by about 0.5 K per day.

The quantification and assessment of the aerosol radiative heating rates utilizing the LASIC campaign data is novel, and is definitely of interest to the scientific community. However, reporting just the radiative heating rates appears to be an underutilization of the modeling tools and observational data that the authors currently use. The manuscript could improve from clearly stating the scientific questions authors want to address to better understand the aerosol-radiation-cloud interactions over SE Atlantic, elaborating on their current findings, and evaluating additional metrics to quantify the aerosol effects on radiation at the TOA and surface, such that these estimates can be easily compared to previous studies over SE Atlantic. I have following comments (both major and minor) and suggestions for edits.

**Major comments:**

1. P3, L9-22: a. The authors claim that "Aerosol impacts on cloud properties resulting in changes in the cloud radiative properties, i.e. aerosol indirect effects, will be captured through the observed cloud properties", yet there is not enough discussion on this topic later on in the results section, especially from the perspective of "indirect effects". One would expect some analysis of the observed cloud microphysical properties to assess the cloud adjustments due to the presence of aerosols. I suggest either removing this sentence from objectives or adding some analysis and discussions to address this topic.

   b. "heating rates are explored along a back trajectory originating at Ascension Island". Please elaborate on the motivation for this part of the study and what scientific questions will this analysis address within section 1.

2. a. Since MERRA-2 thermodynamic profiles are used as inputs to RRTM for heating rate calculations along back trajectories, it would be nice to see a comparison of these variables at least over Ascension Island, where observations are available, to get some sense on representativeness of MERRA-2 thermodynamic profiles compared to the observations from AMF1 or INTERPOSONDE profiles. This is important because at several places within the manuscript, authors bring up anomalous behavior of MERRA-2, with deeper boundary layer, deficiencies in RH profiles, without actually showing comparisons with the observations.

   b. Similarly, even though AOD from MERRA-2 are readily compared to AERONET and AMF1 observations in this study, which is a column integrated and assimilated property

within MERRA-2, some comparisons of aerosol vertical structure, probably using lidar observations from LASIC or other co-located campaigns during this time would be more insightful. P6, L31 mentions that "in agreement with Zuidema et al. (2018), the black and organic carbon in MERRA-2 is located above the cloud layer, but perhaps extends higher in the atmosphere than indicated by lidar observations." Can the authors please clarify which Figure within the specified reference are they alluding to?

3. The authors mention some recent modeling studies, e.g. Chang and Christopher (2017) that used similar techniques/modeling tools as the authors to estimate the aerosol radiative heating rates, as well as direct radiative effects (DREs) of absorbing aerosols at the TOA and surface over SE Atlantic. Therefore, this study could benefit from calculating these additional estimates of DREs at TOA and surface, such that they can compare and contrast the differences in estimates based on the differences in assumptions of aerosol properties, clouds and thermodynamic profiles, as well as the location within SE Atlantic of the current study versus the previous studies.

**Minor/Editorial comments:**

P2, L15: 'lofted to between 3.5 to 4.5 km': Please verify that these heights are above ground level. Also, use of the phrase 'lofted to between' seems inappropriate. Within the boundary layer smoke is well mixed, so to put it more appropriately, 'smoke aerosols extend up to 3.5-4.5 km above ground level'.

P2, L18: 'When compared to satellite observations, models commonly allow for the biomass burning aerosol to descend too rapidly once over the ocean': This applies more to the 'global models' rather than generalizing it to all models.

P13, L9: 'impact of clouds, aerosols, and black carbon': black carbon is part of aerosols, I suggest rewording to mean all aerosols except black carbon and black carbon.

P4, L 9: ice and liquid/ice cloud droplet effective 'radius'?

P4, L 22: vertical profile of aerosols and their 'column' integrated properties?

P5, L14: 'INTERPSONDE profiles were interpolated onto the MERRA-2 vertical profile'? Replace MERRA-2 vertical 'profile' with 'levels'.

P5, L23: The model experiments need elaborate description, may be also tabulation for quick remembering. The authors need to clarify how are clear and cloudy sky cases being simulated, using what classification criteria.

P7, L 5: 'Based on prior results for the height of the aerosol plume, the parcel originated at a height of 2 km.' Please clarify, what prior results are being referred to here? Moreover, this whole paragraph is hard to follow at times, I suggest overhauling and elaborating on how "determining the origin" of aerosol plumes impacts your findings of this study.

P7, L 25 onwards: This paragraph is describing the typical MBL and cloud structure over Ascension, but it appears like a commentary on general cloud features one would observe over this region, rather than depicting these features using the observation data. Moreover, references backing these statements about cloud structure and transitioning lack appropriate referencing.

P7, L 31: authors mention, "bottom panel of Figure 4, which exhibits a subtle, intermittent sub-layer at ~900 hPa". It is hard to make out any intermittent sub-layer at 900 hPa, probably color scale of the figure needs to be improved.

P9, L 1-7: This paragraph is really hard to follow. Authors mention, "heating due to clouds, generally located below 900 hPa, is underwhelming and of similar order of magnitude as the heating due to aerosol" and refer to Figure 6d. From my understanding of Fig. 6, these depict SW heating rates due to aerosols, so I don't understand how are "heating due to clouds" are being inferred.

P9, L 6-7: "in the presence of clouds, radiative heating within the aerosol layer is embellished". Suggest rewording "embellished", as well as clarification on what do the authors mean by this term?

P11, L 20: "It is known that the boundary layer is too deep over Ascension Island in MERRA-2". How is it known, please clarify or use an appropriate reference?

P11, L 22: "SW heating due to aerosol is no longer maximized within the aerosol layer but rather at the surface". Please elaborate why would that be, it is not clear from the current discussions.

**Figures/Tables:**

Table 1: caption says, "Italicized values in parentheses for all aerosols are results with the decreased SSA." I do not see any italicized values in parentheses within the table. Please clarify. Also, consider spelling out M2 to MERRA-2 or explain in caption.

In general, curtain/contour plots are okay, but some sort of mean vertical profiles as line plots are required for understanding the subtle features that the manuscript points to at various instances (e.g. discussions under section 3.2)

Figure 4: Color scale needs changing, as contours are hard to distinguish. Also, can the Y-axis be limited to 400-500 hPa, so that details of the lower troposphere can be highlighted, where the interests of this study lie?

Figure 5: Figure 5b is never discussed, while 5a is barely mentioned. Either remove the figures or include discussions within the main text.

---

## Editor Comment (EC1) · Paquita Zuidema (Editor) · 9 Apr 2020

Title: Radiative Heating Rate Profiles over the Southeast Atlantic Ocean during the 2016 and 2017 Biomass Burning Seasons
Author(s): Allison B. Marquardt Collow, Mark A. Miller, Lynne C. Trabachino, Michael P. Jensen, and Meng Wang
MS No.: acp-2020-106
MS Type: Research article
Iteration: Initial Submission
Special Issue: New observations and related modelling studies of the aerosol–cloud–climate system in the Southeast Atlantic and southern Africa regions (ACP/AMT inter-journal SI)

Overall

The authors present a topic of great interest and importance. The authors present simulations of heating rates due to aerosol layers under clear sky and cloudy conditions over Ascension Island and discuss whether these are representative for the entire South-East Atlantic domain.

Major comments

Overall, the analysis over Ascension Island seemed adequate, although the final assessment that BC is responsible for most of the SW absorption in this location is a bit of a stretch. This is due to the fact that the calculations heavily depend on SSA values, as shown in figure 6 of the manuscript and Table 1. For example, comparing the values contributed by BC only from Fig. 7 to the values in Fig. 6 depends on the SSA assumptions used, where no correction/RH corrected values indeed will give the impression that BC contributes the majority to the heating rate, while if using SSA adjusted for BrC absorption makes BC contribution about 25% (at least by comparing the color scales of the two plots). This is also stressed in their text (and contradicts their conclusions): in lines 15-20 page 9.

There is no conclusion to which of the heating rates calculation in Fig. 6 is the closest one to reality, which might affect the final conclusions. Maybe there are some days where the British CLARIFY aircraft had valid profiles that can support one of these assumptions.

Moreover, the heating rate calculations along the 7-day trajectory from Ascension are not fully clear for some reasons: (1) the calculation procedure is not clear, e.g. does the profiles were taken per each lat/lon along each of the 27 ensemble or whether there was one profile compiled per trajectory. If the latter is correct, then further explanations on the calculation and assumptions is needed. (2) the SSA values selected/assumed over the SEA Ocean, in compared with the values assumed for AI need further elaboration. The authors first claim that MERRA-2 and AI SSA values do not match well, but thereafter claim that over the SEA Ocean they do match (following Shinozuka et al., 2019 analysis). Indeed, in the lower FT it seems that the GOES model (underlying MERRA-2) is able to simulate SSA well (although the current paper talks about 0.92 for SSA over AI, where over the SEA GOES is withing 0.80-0.86 in the lower FT according to Shinozuka et al., 2019, however is underestimating in the mid-FT. The question is which SSA was used then for the vertical profile calculations? Also, it would be of great help to the reader to state the SSA values, both for AI and their MERRA-2 compared values and over the SEA Ocean, since trying to understand which MERRA-2 values compared well with which location was a bit difficult. I am not sure how the lower FT SSA values over the ocean are different than AI values for MERRA-2 and why.

Also, the paper is a bit hard to follow and would benefit from additional editing.

Minor comments

Page 4, line 6, Cimel and not Cimen

Page 4, line 9, cloud effective radii (radii is missing)

Page 6, lines 10-12, why AMF1 and Aeronet Cimels are so different?

Fig. 2, reduce x-axis font

Page 6, line 24, are there evidence of volcanic dust (in the form of size distribution, AE etc.?) during some of the days?

Page 7, lines 8-10, August 2016 (Fig. 3a) shows some contribution from the west, over the ocean as well as from the continent.

Page 8, line 5, please state which observations you are referring to.

Page 8, line 20, aerosols in the (in is missing)

Fig. 6 and 7 might benefit from a similar colorbar (same max-min values) or maybe a plot that shows the accumulating percentages of BC and the other aerosol to the total might be clearer here?

Page 9, lines 8-9, it is unclear why the relative humidity scaled MERRA-2 values were chosen here and not the BrC scaled one?

Table 1, there are no italicized values in parenthesis?

Page 10, lines 3-4, please rephrase

Page 11, lines 7 and onward: please elaborate on the heating rate calculations for the trajectory analysis; as stated above, this is unclear.

Page 12, line 20, the conclusion here contradicts the statement in page 9, lines 15-16.

---

## Editor Comment (EC2) · Paquita Zuidema (Editor) · 9 Apr 2020

To the authors/general public: the 'Editor Comment' is actually from an anonymous referee who was having trouble uploading their review. I thought I could do it anonymously but because I was logged in in 'editor' mode my plan was foiled. Just be aware the comment did not originate from me.

---

## Author Comment (AC1) · 15 May 2020

**Review of "Radiative Heating Rate Profiles over the Southeast Atlantic Ocean during the 2016 and 2017 Biomass Burning Seasons" by Collow et al.**

This manuscript presents a quantitative assessment of the radiative heating rates due to aerosols over Ascension Island based on idealized calculations through a Radiative Transfer Model, using aerosol vertical profiles and optical properties from MERRA-2 reanalysis, thermodynamic profiles, and low-cloud properties from island-based observations during the LASIC field campaign. The authors find shortwave heating within the aerosol layer above Ascension can locally range between 2 and 8 K per day, and shortwave heating due to biomass burning aerosols is not balanced by additional longwave cooling.

The presented assessment of the aerosol radiative heating rates is novel and nicely conducted, which could be informative and insightful to the scientific community if the following concerns and questions are properly addressed and justified. In addition, I find myself having hard times understanding/digesting many of the discussions in the current form of the manuscript. These confusing discussions should be reconstructed and extended with additional details and elaborations before this manuscript can be accepted for publication.

**Major concerns/questions:**

1. The current introduction section seems a little weak in terms of scientific motivations. I recommend stating a stronger scientific motivation for the study, clarifying the scientific question you want to address with this study, and elaborating more on how is your study going to help us better understand the aerosol-radiation-cloud interactions within the SE Atlantic, rather than saying that the goal is to quantify and report the radiative heating rates over Ascension.

The introduction section has been modified to help clarify the motivation behind the work.

2. You compared vertically integrated aerosol properties from MERRA-2 to LASIC measurements, but not thermodynamical properties, especially thermodynamic profiles, which could have subtle impact on the vertical distribution of biomass burning aerosols.

    a. You mentioned in Line 20 on Page 11 that "boundary layer is too deep over Ascension Island in MERRA-2," comparing to LASIC measurements? Could you show the thermodynamic profiles comparison over Ascension?

    b. You mentioned potential deficiencies in RH profiles of MERRA-2, can this also be shown as a comparison with LASIC measurements?

Thermodynamic profiles from MERRA-2 are now included in Figures 5 and 6. A discussion on the comparison between the observations and MERRA-2 has been added to the text.

3. You introduced six sets of experiments towards the end of Section 2. Please specify how are they defined/selected?

Additional text and a table have been added to help clarify the sets of experiments.

    a. Are you using cloud observations to screen for clear skies?

No, cloud observations are not used to screen for clear skies (see response for comment 3b below).

b. Are you simply setting cloud parameter off in the model for clear sky case, even though the inputting thermodynamic profiles felt the existence of a cloud layer? If this is the case, please discuss how is this artifact affecting your results (i.e. is your heating rates under/over-estimating?).

Clear sky here is simply turning clouds off in the model, and you are correct that the thermodynamic profiles will still reflect the existence of a cloud. This is now noted in the text and the implications are discussed.

c. Not all of these experiments are presented in the following results part, e.g. the clean- cloudy case is not shown.

Actually, the clean-cloudy case is indirectly used for the results presented in panels d-f of figure 6. The calculation shown is Cloudy with aerosol – Clean cloudy. This has been made clearer in the text and Table 2.

4. Section 3 is named as "Results," even though 3.1 and 3.2 are not actual "results" of the radiative transfer calculations. Using "Results" as a section name is vague, and I suggest reconsidering the section organization, one option could be making 3.1 as a new Section 4, and 3.2 as a new Section 5. Currently, all your results are lumped into one section (3.3), which is poorly organized, I suggest making 3.3 as a new Section 6, and break each part into subsections, e.g. SSA sensitivity, SW heating due to black carbon, SW heating enhancement in the presence of clouds, LW cooling, and heating rates along back-trajectories.

The suggested section organization has been implemented.

5. I am a bit lost regarding the purpose of discussing the back-trajectories, how are they (or the origin of aerosol) related to (or affect?) your heating rate calculations? Besides, some of the details regarding the HYSPLIT runs should be introduced in the Data & Methodology section. P7 L15-18 seem abrupt at the end of Section 3.1, they also fit better in the Data & Methodology section. Perhaps these last two paragraphs of Section 3.1 could be moved to Section 2 together.

The back trajectories have been included as they impact the heating rates of the aerosol as the plume approaches Ascension Island. A sentence has been added to the text regarding this. The details on the runs have been moved to the Data and Methodology section as suggested.

a. P7 L2, "determine the origin of the aerosol" is a strong statement, as back-trajectories not necessarily indicate the exact pathways of biomass burning plumes, and one has to combine other sources, e.g. fire emission data, in order to determine the origin of the aerosol observed at Ascension. I suggest rewording.

This sentence has been reworded to indicate we are interested in the trajectory of the aerosol.

b. P7 L5-6, there were aerosol both in the free-troposphere and boundary layer, why 2 km is picked? What is the prior results you are referring to, please provide the reference.

The height of 2 km was selected as that is the middle of the aerosol layer detected by the micropulse lidar. This can also be seen in Figure 4 of Zuidema et al. (2018), which is now cited in the text.

c. P7 L9-10, why the subtropical high over the southern Indian Ocean plays a role here? Monthly mean SLPs could be added to the background of Fig. 3 to base this statement.

The southern Indian Ocean is too far east to nicely include in the figure, but adding monthly mean SLP to the background was a great idea and has been incorporated.

d. P7 L10-14, what are these observations implicating, how are they related to this work, or affecting the interpretation of your results?

The back trajectories have implications on the AOD over the site, as well as the heating within the aerosol layer as it travels towards Ascension Island. Given that this is controlled by the large-scale circulation, the amount of heating on days (and years) not shown can be extrapolated. A sentence regarding this has been added to the text.

6. Regarding Fig. 4, although one can identify an inversion layer from temperature profiles (distinguishing yellow and light orange from your plot), I would argue that potential temperature is a better choice to show the cloud top inversions. I think you could also extend the discussion to the potential role of the RH plot on indicating the biomass burning smoke plumes arriving at Ascension, as RH bursts in the free-troposphere tend to colocate well with the smoke episodes arriving at Ascension.

We investigated showing potential temperature instead however the figures did not prove to show cloud top inversion any better than temperature. As a result, we had elected to stick with temperature for the figure. A discussion on the connection between the smoke plumes and relative humidity has been added to the text.

7. You did not discuss Figure 5b at all, and only mentioning Fig. 5a for clouds that are not visible in the plot. I wonder if this figure is necessary, it doesn't seem to add much useful information, and you barely discussed it in the main text.

The fields shown in Figure 5 are used as an input to the radiation transfer calculations. This is also the first time (to our knowledge) that the microphysical properties of clouds during LASIC have been included in peer reviewed literature. The section of the text has been expanded.

8. The discussion in the first paragraph of Page 9 is very confusing to me. In the first couple of sentences, you mentioned comparing heating due to clouds with heating due to aerosols, and my interpretation of Figure 6d is that this is SW heating due to aerosol under a cloudy sky, and you did not show a case for heating due to clouds alone (no aerosol), so how did you make the comparison? whereas in the last sentence you talked about comparing aerosol heating under cloudy and clear skies (isn't this contradicting to the first couple of sentence? please clarify), and what do you mean by "embellished," I had trouble relating this word to the observations.

This paragraph has been rewritten, with an additional figure and references to the table added. "enhanced" is now used instead of "embellished".

9. I also think the discussion for Figure 6d,e,f could be substantially extended. Currently, you barely discussed them ("embellished" is all you used to describe the comparison), and I will be curious to know why SW heating due to aerosol in the BL is enhanced under cloudy conditions? This is contradicting to my intuition, as the cloud layer reflects SW back to the space, I would expect the SW heating in the BL to decrease instead of increase.

This paragraph has been rewritten and expanded. While clouds reflect SW back to space, the radiation encounters the aerosol layer above the cloud layer before it can leave at the top of the atmosphere. This means that within the aerosol layer, there are two opportunities for absorption.

10. It seems to me that MERRA-2 is not distributing enough BC/OC in the BL based on your Figure 2, 6 and 7 (clear sky condition). I wonder if you have compared MERRA-2 aerosol vertical profiles with extinction profiles from LASIC MPL (when available) or NASA ORACLES HSRL2 profiles or UK CLARIFY EXCALABAR profiles when they become available over Ascension during the 2017 season?

A figure showing backscatter from the LASIC MPL has been added to assist in an evaluation of the vertical profile of aerosol. MERRA-2 actually has more extinction from aerosol in the boundary layer than aloft, however this is almost completely a result of sea salt.

11. Regarding the LW cooling associated with the SW heating, the concern I have is that the observed temperature profiles (from LASIC radiosondes) had already felt that heating, in other words, the temperature increase was already taken into account in the observed profiles. By modifying the observed T profiles, you're artificially increasing the temperatures (artificially boosting the LW cooling). I would recommend just simply turn on the LW calculation using the same observed T profiles, and see if the net radiation budget produce a cooling or a warming.

We agree that by adding the heating rate to the temperature, the temperature profiles had already experienced heating due to aerosols. Accordingly, we have now reworked the LW calculation to account for this.

12. You should state whether this LW experiment is calculated with cloud presence or not.

The calculation was done without clouds which is now noted in the text.

13. The discussion in lines 17-32 on page 10 is particularly hard for me to digest. As we know LW cooling is always happening no matter aerosol presents or not, and the "LW cooling" you are talking about in this paragraph is the additional LW cooling caused by the increase in the temperature profiles due to SW heating (since it is done by subtracting a control run). The following points should be addressed properly in order to make the discussion clear.

      a. When you say "LW radiational cooling never offsets the absorption due to aerosols," you should make this clear that you mean the additional LW cooling never offsets the absorption.

This has been clarified.

b. L 25, "magnitude of the LW cooling never reaches ..." same problem as above. LW cooling at inversions can easily reaches 10K/day at night. The LW cooling you are referring to is the difference between the T-modified run and the control run. Please make this clear.

This has been made clearer.

c. L 23, "radiational cooling still occurs..." As mentioned above, LW radiative cooling always occurs no matter the aerosol condition. Since you are showing the difference between the T-modified run and the control run, as long as there is SW heating due to aerosol (no matter how much), T profile will be modified, and difference in LW heating will exist. This cannot be used to demonstrate that additional heating due to aerosol remains in the column, you have to use the real LW heating profile to quantify that, not the difference between two runs. I strongly recommend re-assessing this LW part (see Major comment 11), at least the way you interpret/discuss it.

These points have been clarified in the text and we have changed the methodology for the LW calculations as suggested by Major comment 11.

14. Regarding your case study on the back trajectory, first, please specify reasons for originating at 2 km, second, why the meteorology for HYSPLIT runs are switched to GDAS instead of MERRA-2 as you did for the monthly back-trajectories, is MERRA-2 not capable to do ensemble runs? Please justify. Then, trajectories were forced by GDAS but radiative transfer calculations were using MERRA-2 thermodynamic profiles (why inconsistent)?

A height of 2 km was selected as the origin of the back trajectories based on the central location of the aerosol layer using micropulse lidar back scatter data. There are no ensembles associated with MERRA-2, whether related to the data assimilation process or the actual analysis. This point has been made clearer in the text. The same case study was carried out using the MERRA-2 back trajectories and has now been added to the figure. Given the uncertainty associated with back trajectories, we wanted to make sure there was a measure of the variability for the heating rates. As a result, that is why we ultimately decided to use GDAS. MERRA-2 is one of the few reanalyses that include the assimilation of aerosol optical depth, making it the obvious choice for aerosol properties that are consistent with a thermodynamic profile.

15. Why is the SW heating along the back trajectory limited to below the inversion? I would expect there to be aerosol in the FT along the 7-day back trajectories, and why not showing the aerosol and thermodynamic curtain plots along the trajectories from MERRA-2? In Lines 18-20, you're saying the aerosol layer is entirely above the inversion along the trajectory, and yet, no heating above the inversion? This is very confusing, please justify.

This comment was extremely helpful. We found a bug in the aerosol input during our investigation, and the figure has been updated accordingly. The results are more reasonable now.

16. In the last paragraph, you mentioned that the ultimate goal is to study how the heating due to aerosols impacts the transition of marine stratocumulus to trade cumulus, I would really love to see more discussions added to the manuscript on how will this study help towards achieving this goal. For instance, how can this study contribute to the understanding of cloud adjustments to aerosols, and what insights can this study provide on the stratocumulus to cumulus transition in

the southeast Atlantic. Such discussions will substantially strengthen the scientific importance of this study.

We have added a paragraph to the conclusions that discusses our results in the context of the stratocumulus to cumulus transition.

**Minor issues:**

*Abstract*

• Line 30, you mentioned "stabilization of the lower troposphere," but this is not discussed anywhere in the main text of manuscript. I suggest adding discussion regarding this point you raised in the abstract.

A discussion on this topic has been added to the conclusion section.

*Introduction*

• P2 L21, you haven't introduced Ascension Island yet, a general reader would have no idea where the island is, near coast? or in the remote ocean? I suggest introducing Ascension Island somewhere in the introduction.

Ascension Island is now described in the introduction on line 21 of page 2.

• P3 L10-15, these information on datasets belong to the Data section, seems to me.

This paragraph has been modified with the information on the datasets getting moved to the Data section.

• P3 L16-18, these sentences seem to belong to the Methodology section. I would suggest adding more motivational statements, clarifying your scientific goals, here in the last paragraph of the introduction, replacing these details of datasets and approaches.

We contemplated moving these sentences but decided to leave them in the current section. As we have noted above based upon your previous comments, we agreed with your assessment about the motivation for the study and have made substantial changes to the manuscript in this regard.

*Data and Methodology*

• P4 L7-, Because of the location of the AMF1 site, orographically generated clouds frequent present in LASIC AMF1 cloud measurements, please address how will this feature affect your assessment and the general representativeness of your results.

We added a paragraph to the paper that discusses our analysis of the island effects on the cloud observed at the AMF1 site. In short, we analyzed the clouds observed during the entire year and found them to have unequivocally originated from the air near the ocean surface. Their cloud bases correspond to the marine LCL rather than the LCL at the AMF1 site. They are almost certainly modified by the orography given that we observe a systematic updraft of on average 0.5 m s$_{-1}$ near the surface at the AMF1 site to near zero at 600 m. Orographically lifting all near-surface marine parcels to the near their LCL substantially increases the probability of cloud

development and we recognize that this enhancement likely impacts our radiation transfer calculations. To adequately address this issue, we note in the revised manuscript that our radiation transfer calculations relative to clouds may serve as an upper bound to these impacts.

Please also specify the temporal resolutions of ARSCL and MICROBASE.

The temporal resolution of 4 seconds for ARSCL and MICROBASE is now noted in the text.

*Results*

*3.1 Evaluation of Aerosols in MERRA-2*

• P6 L23, my understanding of the location of AMF1 site is that it is elevated and located at the upwind part of the island, which should be representative of the aerosol condition of a marine boundary layer (minimal island effect). Besides, if indeed there were dust (more scattering) mixed into the AMF1 sampling volume, shouldn't we expect a higher SSA? (Zuidema et al. 2018b's values are lower). Please correct me if this is not the case.

The aeronet site is actually located by the airport, on the opposite side of the island from where the AMF1 was located. We have removed the statement regarding volcanic dust since we cannot prove that is the case.

• P6 L28-29, could be helpful if AODs are overlaid on top of Fig. 2.

This made the figure look too busy but the AODs are now including in the heating rate figures.

• P6 L34, the decrease in BL height is not very evident based on Fig. 2, overlaying some other forms of indication could be helpful.

We have removed to statement regarding the BL height. A decrease in BL height was more obvious in a previous version of the figure that included October as well August and September.

*3.2 Thermodynamic Profiles over Ascension Island*

• P7 L21, "time-height" should be time-pressure, as you're showing pressure in the vertical.

This has been corrected.

• P7 L22, cloud top inversions at Ascension are not around 700 hPa (~3 km). Please double check the pressure axes in Figure 4.

Yes, you are correct. This was a plotting issue that has been resolved.

• P7 L31, I do not see a "subtle, intermittent sub-layer at ~900 hPa" based on the RH curtain plot, perhaps this will be more visible in a single-profile presentation. Based on Fig. 13 of Zhang and Zuidema (2019), the intermittent layer seems to be located at ~700 m, which is lower than 900 hPa.

A figure has been added to show the monthly mean vertical profile of temperature and RH.

*3.3 Heating Rate Profiles over Ascension Island*

• P8 L21, it would be easier to visualize this co-variability between the heating rate and the AOD, if you could add MERRA-2 aerosol contours or AOD time series in the background of Fig. 6.

AOD contours have been added to the background of the figure.

• P8 L29, if it is hard to tell with the color bar, could you provide some values to indicate the difference between heating rates calculated from MERRA-2 SSA and RH scaled SSA?

The reader is now referred to Table 3 which shows the heating rates with the different SSAs.

•P9 L18-19, I see heating due to black carbon ~0.5 K/day extending to around 600 hPa just as in Figure 6, please re-state your argument about this observation. Besides, how do you know it is absorption from dust (isn't dust more scattering)? Please justify.

This paragraph has been rewritten with a figure than now shows the percentage of heating due to aerosols that is from black carbon.

• P9 L29, please define "enhancement of heating within the aerosol layer due to clouds" in the text or in the caption, i.e. how did you quantify that, is this cloudy-aerosol run minus clear-aerosol run (Fig. 6f – Fig. 6c = Fig. 8a?)?

A table has been added to clarify the quantities that are shown in the figures.

• P9 L30-32, "a few K per day"? the color bar on Fig. 8 only goes to 0.6, how did you get a few K per day? "...but when all aerosols are considered the majority of the enhancement is located ..." isn't this true for both 'All Aerosols' and 'Black Carbon'? Please check your logic here.

This should have been a few *tenths of a* K and has been corrected.

• P9 L32-34, could you please extend this discussion, especially on why this BL enhancement is not apparent for Black Carbon only case, even though BC is highly absorptive?

There is a minimal amount of black carbon in the boundary layer, which is now noted in the text.

• P10 L1-2, I think you should be careful here and say "...due to the presence of clouds..."

This has been modified as suggested.

• P10 L9 and thereafter, you used the phrase "radiational cooling," while I am more used to seeing "radiative cooling" being used in other literatures.

*Radiative* cooling is now used throughout.

• P10 L28-29, "some heating occurs above and below the aerosol," we can't tell where the aerosol layer is based on this plot, one option is to put MERRA-2 aerosol contours in the background. Another option is to show a line plot highlighting a single heating profile along with the aerosol profile.

Contours of AOD have been added to the figures.

• P10 L29, please discuss how is this redistribution of heat, as you put it, going to modify the stability of the boundary layer, as you pointed out in the abstract.

A discussion has been added to the conclusion section on this topic.

• P11 L21, please elaborate more on how the depth of BL affects the SW heating. Why minimal SW heating occurs in the last few hours of the back trajectory?

This is not present in the updated version of the figure following updates to the calculation.

• P11 L23, please explain or discuss your speculations on why SW heating maximized at the surface here.

This statement no longer applies to the updated figure.

*Summary and Conclusions*

• P12 L12, "...greater depth of the boundary layer..." comparing to what?

This statement is not in the revised version.

• P12 L16, "local heating rates are sensitive to the thickness of the aerosol plume" this is not discussed in the results section. Which figure supports this argument? Please mention this argument when you discuss that figure.

This is discussed as part of Figure 8, which is now noted in the text.

• P12 L20, "...most of the SW absorption" please be quantitative here.

It is now noted that up to 80% of the SW absorption can be due to black carbon.

• P12 L21, please be specific about this statement, i.e. which month? over Ascension or the whole SE Atlantic? Zuidema et al. 2018b states smoke often presents in the BL of Ascension Island, more frequent than "at times."

Despite the title of Zuidema et al. (2018), the conclusions presented in the paper are in good agreement with our use of "at times" for the months of August and September as there is a periodicity in aerosol concentrations. We have specified that we are referring to the months of August and September.

From Zuidema et al. (2018):

"1. Near-surface rBC mass concentrations vary significantly at synoptic time scales from June to October at the Ascension Island location….

3. The aerosol loadings within and above the cloudy boundary layer do not necessarily correlate well, with more of the total column aerosol present in the boundary layer early in the BBA season, migrating to predominantly free-tropospheric aerosol in September."

• P12 L23-24, I think what you want to express here is that adding a cloud layer will result in an enhancement of heating. Saying "interaction between SW radiation, clouds, and aerosols" is a bit misleading, as aerosols and clouds are not interacting in your calculations. I suggest a more careful rewording.

This sentence has been reworded.

• P12 L29, I didn't think you were trying to represent the entire southeast Atlantic using Ascension observations until I saw this statement, and I don't think this study should be used to represent the entire region. I suggest stating this clearly in the introduction or data section, that this study only represents the remote SE Atlantic, and cannot be used to represent the entire region. Also, you could change the title to "...remote SE Atlantic..."

This statement has been removed.

• P12 L31, "sensitive the heating ... is," sensitive to what?

This has been clarified to refer to the aerosol optical properties.

•P13 L15-20, the first and third sentences are the same sentence, please double check.

This has been fixed.

*Figure/Table issues*

1. Table 1, there are no italicized values in this table, please check.

Yes, that caption was for an older version of the table and has since been corrected.

2. Figure 2, please use 10 -5 instead of e-05

This has been modified as suggested.

3. Figure 4, please double check pressure axis, you are showing a 3 km BL.

The pressure axis has been fixed.

4. Figure 5, in my opinion, this figure can be removed, and all the white space above 800 hPa can be minimized.

Rather than removing Figure 5, the discussion pertaining to this figure has been extended. The y axis has been modified to only show below 800 hPa.

5. Figure 6, SSA instead of "SSA albedo." Again, the space above 600 hPa can be minimized, same for Fig. 7, 8 and 9.

This has been fixed.

6. I think for Figure 7 and after, you probably should remind the reader that we should compare these results only with the bottom panel of Fig. 6 (the RH scaled one), by making a note in the caption that SSA is the RH scaled one.

This is now noted in the figure captions as suggested.

7. Most of the results are presented in curtain plots, they are nice in terms of showing the whole month, but rather poorly representing details in vertical. I recommend showing couple plots with single profiles when you discuss details in vertical, especially when you discuss the relative location of heating/cooling to the aerosol layer.

A couple figures have been added that show the mean profile for the month for the thermodynamics and radiative heating.

---

## Author Comment (AC2) · 15 May 2020

Review on "Radiative Heating Rate Profiles over the Southeast Atlantic Ocean during the 2016 and 2017 Biomass Burning Seasons" by Collow et al.

This study attempts to quantify the contribution of aerosols and clouds on radiative heating rates within the atmospheric column over Ascension Island in South-east (SE) Atlantic. The approach involves the use of thermodynamic profiles and low-cloud observations during LASIC field campaign and aerosol profiles from MERRA-2 reanalysis data as inputs to a Radiative Transfer Model (RRTM). The study finds that on average, the maximum local aerosol SW heating within the column over the course of the biomass burning season ranges from 2 to 4 K per day. In addition, on days biomass burning aerosol plumes are observed above clouds, shortwave heating within the aerosol plume is enhanced by about 0.5 K per day.

The quantification and assessment of the aerosol radiative heating rates utilizing the LASIC campaign data is novel, and is definitely of interest to the scientific community. However, reporting just the radiative heating rates appears to be an underutilization of the modeling tools and observational data that the authors currently use. The manuscript could improve from clearly stating the scientific questions authors want to address to better understand the aerosol-radiation-cloud interactions over SE Atlantic, elaborating on their current findings, and evaluating additional metrics to quantify the aerosol effects on radiation at the TOA and surface, such that these estimates can be easily compared to previous studies over SE Atlantic.

I have following comments (both major and minor) and suggestions for edits.

We sincerely thank the reviewer for the constructive feedback.

Major comments:
1. P3, L9-22: a. The authors claim that "Aerosol impacts on cloud properties resulting in changes in the cloud radiative properties, i.e. aerosol indirect effects, will be captured through the observed cloud properties", yet there is not enough discussion on this topic later on in the results section, especially from the perspective of "indirect effects". One would expect some analysis of the observed cloud microphysical properties to assess the cloud adjustments due to the presence of aerosols. I suggest either removing this sentence from objectives or adding some analysis and discussions to address this topic.
The reference to aerosol indirect effects has been removed.

b. "heating rates are explored along a back trajectory originating at Ascension Island". Please elaborate on the motivation for this part of the study, and what scientific questions will this analysis address within section 1.
The introduction and motivation for the study have been modified to reflect that we are quantifying uncertainties associated with aspects of the radiative heating due to aerosols.

2.a. Since MERRA-2 thermodynamic profiles are used as inputs to RRTM for heating rate calculations along back trajectories, it would be nice to see a comparison of these variables at least over Ascension Island, where observations are available, to get some sense on representativeness of MERRA-2 thermodynamic profiles compared to the observations from AMF1 or INTERPSONDE profiles. This is important because at several places within the manuscript, authors bring up anomalous behavior of MERRA-2, with deeper boundary layer, deficiencies in RH profiles, without actually showing comparisons with the observations.

MERRA-2 thermodynamic profiles are now included and evaluated against the interpsonde observations.

b. Similarly, even though AOD from MERRA-2 are readily compared to AERONET and AMF1 observations in this study, which is a column integrated and assimilated property within MERRA-2, some comparisons of aerosol vertical structure, probably using lidar observations from LASIC or other co-located campaigns during this time would be more insightful. P6, L31 mentions that "in agreement with Zuidema et al. (2018), the black and organic carbon in MERRA-2 is located above the cloud layer, but perhaps extends higher in the atmosphere than indicated by lidar observations." Can the authors please clarify which Figure within the specified reference are they alluding to?

A figure has been added showing the backscatter from the MPL to compare to the MERRA-2 vertical profile, in addition to a reference to Figure 4 of Zuidema et al. (2018).

3. The authors mention some recent modeling studies, e.g. Chang and Christopher (2017) that used similar techniques/modeling tools as the authors to estimate the aerosol radiative heating rates, as well as direct radiative effects (DREs) of absorbing aerosols at the TOA and surface over SE Atlantic. Therefore, this study could benefit from calculating these additional estimates of DREs at TOA and surface, such that they can compare and contrast the differences in estimates based on the differences in assumptions of aerosol properties, clouds and thermodynamic profiles, as well as the location within SE Atlantic of the current study versus the previous studies.
A figure (and associated text) has been added to show the DRE at the TOA and surface, with a couple sentences comparing to Chang and Christopher (2017).

Minor/Editorial comments:
P2, L15: 'lofted to between 3.5 to 4.5 km': Please verify that these heights are above ground level. Also, use of the phrase 'lofted to between' seems inappropriate. Within the boundary layer smoke is well mixed, so to put it more appropriately, 'smoke aerosols extend up to 3.5 -4.5 km above ground level'.
Yes, this is above ground level. The sentence has been modified as suggested.

P2, L18: 'When compared to satellite observations, models commonly allow for the biomass burning aerosol to descend too rapidly once over the ocean': This applies more to the 'global models' rather than generalizing it to all models.

"global" has been added to this sentence.

P3, L9: 'impact of clouds, aerosols, and black carbon': black carbon is part of aerosols, I suggest rewording to mean all aerosols except black carbon and black carbon.

This sentence has been reworded.

P4, L 9: ice and liquid/ice cloud droplet effective 'radius'?

Yes, "radius" has been added to the sentence.

P4, L 22: vertical profile of aerosols and their 'column' integrated properties?
"Column" has been added to the sentence.

P5, L14: 'INTERPSONDE profiles were interpolated onto the MERRA-2 vertical profile'?
Yes, this was done so that RRTM could be run using the MERRA-2 vertical profile of aerosols which is now noted in the text.

Replace MERRA-2 vertical 'profile' with 'levels'.
"Profile" has been replaced by "levels".

P5, L23: The model experiments need elaborate description, may be also tabulation for quick remembering. The authors need to clarify how are clear and cloudy sky cases being simulated, using what classification criteria.
Additional text has been added to this paragraph as well as a table to help clarify the experiments.

P7, L5: 'Based on prior results for the height of the aerosol plume, the parcel originated at a height of 2 km.' Please clarify, what prior results are being referred to here? Moreover, this whole paragraph is hard to follow at times, I suggest overhauling and elaborating on how "determining the origin" of aerosol plumes impacts your findings of this study.
"Prior" results have been clarified. A sentence has also been added to show the relevance of the back trajectories for the study. Note, this paragraph is now located in Section 2.4.

P7, L 25 onwards: This paragraph is describing the typical MBL and cloud structure over Ascension, but it appears like a commentary on general cloud features one would observe over this region, rather than depicting these features using the observation data. Moreover, references backing these statements about cloud structure and transitioning lack appropriate referencing.
This paragraph is now in better connection to the figure, with an expansion of the discussion on observed cloud properties.

P7, L 31: authors mention, "bottom panel of Figure 4, which exhibits a subtle, intermittent sub-layer at ~900 hPa". It is hard to make out any intermittent sub-layer at 900 hPa, probably color scale of the figure needs to be improved.

This can now easily be seen in the figure that was added showing the month averaged vertical profile (Figure 5 in the revised text).

P9, L 1-7: This paragraph is really hard to follow. Authors mention, "heating due to clouds, generally located below 900 hPa, is underwhelming and of similar order of magnitude as the heating due to aerosol" and refer to Figure 6d. From my understanding of Fig. 6, these depict SW heating rates due to aerosols, so I don't understand how are "heating due to clouds" are being inferred.
This paragraph has been edited, with references for the heating due to clouds added for the table with the heating rates.

P9, L 6-7: "in the presence of clouds, radiative heating within the aerosol layer is embellished". Suggest rewording "embellished", as well as clarification on what do the authors mean by this term?

The word "enhanced" is now used.

P11, L 20: "It is known that the boundary layer is too deep over Ascension Island in MERRA-2". How is it known, please clarify or use an appropriate reference?
This sentence has been since been removed.
P11, L 22: "SW heating due to aerosol is no longer maximized within the aerosol layer but rather at the surface". Please elaborate why would that be, it is not clear from the current discussions.
This paragraph has been rewritten as a bug was found in the radiation transfer aerosol input. This has since been corrected and the results are now more reasonable.

Figures/Tables:
Table 1: caption says, "Italicized values in parentheses for all aerosols are results with the decreased SSA." I do not see any italicized values in parentheses within the table. Please clarify. Also, consider spelling out M2 to MERRA-2 or explain in caption.
This was from an earlier version of the table and has been removed.

In general, curtain/contour plots are okay, but some sort of mean vertical profiles as line plots are required for understanding the subtle features that the manuscript points to at various instances (e.g. discussions under section 3.2)
A mean vertical profile has been added as suggested (Figure 9 in the revised text).

Figure 4: Color scale needs changing, as contours are hard to distinguish. Also, can the Y-axis be limited to 400-500 hPa, so that details of the lower troposphere can be highlighted, where the interests of this study lie?
Figure 4 (Figure 6 in the revised text) has been modified. The contours are now easier to distinguish and the y axis has been limited to 400 hPa.

Figure 5: Figure 5b is never discussed, while 5a is barely mentioned. Either remove the figures or include discussions within the main text.

The text associated with Figure 5 (Figure 7 in the revised text) has been expanded upon.

---

## Author Comment (AC3) · 15 May 2020

Title: Radiative Heating Rate Profiles over the Southeast Atlantic Ocean during the 2016 and 2017 Biomass Burning Seasons
Author(s): Allison B. Marquardt Collow, Mark A. Miller, Lynne C. Trabachino, Michael P. Jensen, and Meng Wang
MS No.: acp-2020-106
MS Type: Research article
Iteration: Initial Submission
Special Issue: New observations and related modelling studies of the aerosol–cloud–climate system in the Southeast Atlantic and southern Africa regions (ACP/AMT inter-journal SI)

Overall The authors present a topic of great interest and importance. The authors present simulations of heating rates due to aerosol layers under clear sky and cloudy conditions over Ascension Island and discuss whether these are representative for the entire South-East Atlantic domain.

We thank the reviewer for the time spent and helpful feedback.

Major comments
Overall, the analysis over Ascension Island seemed adequate, although the final assessment that BC is responsible for most of the SW absorption in this location is a bit of a stretch. This is due to the fact that the calculations heavily depend on SSA values, as shown in figure 6 of the manuscript and Table 1. For example, comparing the values contributed by BC only from Fig. 7 to the values in Fig. 6 depends on the SSA assumptions used, where no correction/RH corrected values indeed will give the impression that BC contributes the majority to the heating rate, while if using SSA adjusted for BrC absorption makes BC contribution about 25% (at least by comparing the color scales of the two plots). This is also stressed in their text (and contradicts their conclusions): in lines 15-20 page 9. There is no conclusion to which of the heating rates calculation in Fig. 6 is the closest one to reality, which might affect the final conclusions. Maybe there are some days where the British CLARIFY aircraft had valid profiles that can support one of these assumptions.
An effort has been made to clarify the conclusions on the contribution of biomass burning aerosol to the SW heating. Unfortunately, we were unable to find data from CLARIFY to validate the results over Ascension Island.

Moreover, the heating rate calculations along the 7-day trajectory from Ascension are not fully clear for some reasons:
(1) the calculation procedure is not clear, e.g. does the profiles were taken per each lat/lon along each of the 27 ensemble or whether there was one profile compiled per trajectory. If the latter is correct, then further explanations on the calculation and assumptions is needed.
The profiles were taken for each lat/lon along the 27 individual trajectories. This has been made clearer in the text.

 (2) the SSA values selected/assumed over the SEA Ocean, in compared with the values assumed for AI need further elaboration. The authors first claim that MERRA-2 and AI SSA values do not match well, but thereafter claim that over the SEA Ocean they do match (following Shinozuka et al., 2019 analysis). Indeed, in the lower FT it seems that the GOES model (underlying MERRA-

2) is able to simulate SSA well (although the current paper talks about 0.92 for SSA over AI, where over the SEA GOES is withing 0.80-0.86 in the lower FT according to Shinozuka et al., 2019, however is underestimating in the mid-FT. The question is which SSA was used then for the vertical profile calculations? Also, it would be of great help to the reader to state the SSA values, both for AI and their MERRA-2 compared values and over the SEA Ocean, since trying to understand which MERRA-2 values compared well with which location was a bit difficult. I am not sure how the lower FT SSA values over the ocean are different than AI values for MERRA-2 and why.

As the aerosol plume is transported across the ocean, in reality, black carbon becomes coated with organic carbon, forming tar balls with optical properties that change the more the aerosol ages. This is not represented in MERRA-2, given the lack of brown carbon. Closer to the African coast, there is less time for the aerosol to age, and therefore, the optical properties are likely more similar to black carbon. By the time the aerosol reaches the island, physical properties of the aerosol have changed considerably. The results demonstrated by Shinozuka et al., 2019 demonstrate this. GEOS had excellent agreement with the ORACLES observations for SSA along the African coast, however, struggled closer to Ascension Island. A sentence has been added to the text reflecting this.

Also, the paper is a bit hard to follow and would benefit from additional editing.
The manuscript has undergone substantial changes that hopefully make it easier to follow.

Minor comments
Page 4, line 6, Cimel and not Cimen
This has been fixed.

Page 4, line 9, cloud effective radii (radii is missing)
This has been fixed.

Page 6, lines 10-12, why AMF1 and Aeronet Cimels are so different?
There are a few reasons the AOD from the AMF1 and Aeronet are not identical. 1) The AMF1 AOD is actually from an MFRSR, not a Cimel sun photometer. 2) The Aeronet site is located at the airport on the eastern side of the island, while the AMF1 was stationed at a higher elevation of the southwestern side of the island. 3) Given the different instruments and institutions (NASA for Aeronet, DOE for AMF1), there are different processing algorithms that were used to generate the AOD, though this is not likely the leading cause for any differences.

Fig. 2, reduce x-axis font
The font size has been reduced as suggested.

Page 6, line 24, are there evidence of volcanic dust (in the form of size distribution, AE etc.?) during some of the days?

We cannot say for sure.  There is no instrument at the AMF1 site that can measure coarse mode aerosols or determine their chemical composition.  There was a Proton Transfer Mass Spectrometer (PTR-MS) deployed at the AMF1 site during LASIC, but data from this PTR-MS have not been processed and there is no plan to do so in the near term according to the

instrument operator. Absorption by volcanic aerosols would be present in the aerosol radiation measurements made by the PSAP (Particle Soot Absorption Photometer) at the AMF1, but there would be no way to separate the absorption from biomass burning aerosol from that volcanic aerosols. The second author of this study calibrated sun photometers on multiple occasions at the Mauna Loa Observatory in Hawaii, which lies at 3400 m in an extensive volcanic field subject to high winds, but no evidence of volcanic aerosol was found in these observations. While we cannot be completely sure that such aerosols are not present at the AMF1 site, we suspect that appreciable concentrations are unlikely given the windward location of the AMF1 and that there is active volcanic activity on Ascension Island.

Page 7, lines 8-10, August 2016 (Fig. 3a) shows some contribution from the west, over the ocean as well as from the continent.
With the exception of one day (which probably has an incorrect trajectory due to errors in the MERRA-2 wind field), the trajectories that originate over the Southeast Atlantic Ocean do pass through the interior of the African continent.

Page 8, line 5, please state which observations you are referring to.
A reference has been added to this line for the observations.

Page 8, line 20, aerosols in the (in is missing)
This sentence has been updated.

Fig. 6 and 7 might benefit from a similar colorbar (same max-min values) or maybe a plot that shows the accumulating percentages of BC and the other aerosol to the total might be clearer here?
We now have a figure that shows the percentage of heating due to black carbon instead.

Page 9, lines 8-9, it is unclear why the relative humidity scaled MERRA-2 values were chosen here and not the BrC scaled one?
The RH scaled SSA was chosen to present the middle of the road scenario, which is now noted in the text. Additionally, SSA observations are only available at the surface. Assumptions had to be made that the vertical profile of SSA was representative of the observations. By using the RH scaled scenario, we have a true observationally based correction.

Table 1, there are no italicized values in parenthesis?
The italicized values were in a previous version of the table however the caption was never updated. This is now fixed.

Page 10, lines 3-4, please rephrase
This sentence has been rewritten.

Page 11, lines 7 and onward: please elaborate on the heating rate calculations for the trajectory analysis; as stated above, this is unclear.
This section has been rewritten.

Page 12, line 20, the conclusion here contradicts the statement in page 9, lines 15-16.

The statements that were on page 9 have been adjusted following the new figure showing the percentage of heating due to black carbon.

---

## Referee Report (RR1)

**Review on revised "Radiative Heating Rate Profiles over the Southeast Atlantic Ocean during the 2016 and 2017 Biomass Burning Seasons" by Collow et al.**

The authors have made a substantial effort in revising the manuscript especially the Introduction section, with stronger motivation and literature review. The organization of sections is also better overall. The authors have made thoughtful responses to the reviewer's comments and incorporated most of the suggested changes. I recommend that this version of the manuscript be published pending some minor comments/technical corrections listed below:

Abstract, Line 30: .... extremely sensitive to the single-scattering albedo assumptions in the models.

Line 15:  biomass burning aerosol plumes/layers extend up to 3- 4.5 km....

Line 18-19: 'global models commonly allow' -> please change the wording here to something like, models simulate.

Line 33: "SSA are from MERRA-2, and were scaled in the vertical by the profile of mixing ratio for the individual species (GMAO, 2015; GMAO, 2015b). The value for SSA at 550 nm from MERRA-2 was used and assumed to be spectrally independent."

Since you mention in your findings that assumptions of SSA are very important for heating rate calculations, could you list a number or a range for SSA values that you calculated in the model for the typical smoke mixture (based on the mixing ratio profiles) you observed in your study? Also, could you add a few sentences on what are the implications on using spectrally 'independent' SSA for you heating rate calculations, and how would they differ from reality?

---

## Editor Decision (ED1)

Dear Authors - Thank you for submitting your manuscript to ACP. I have reviewed the referees' comments and your responses to them, and have a few additional comments I would like you to consider before this manuscript goes to publication. The manuscript will not go back to the referees again. The page and line numbers of the comments are based on the Author's Response file you provided. It doesn't look to me that that shows the tracked changes to the original manuscript, as requested by the journal, but I can live with that. You may follow suit with your next revision but I would like to see your responses indicated point-by-point to the comments below.

Sincerely, Paquita Zuidema

Abstract, line 18: "this has yet to be quantified" is a bit strong. Radiative heating profiles have been calculated before, but not specifically for the air masses you are targeting. Please consider revising this sentence.

Abstract: My understanding is that acronyms should not be used or defined within the abstract unless substantially used. I only see SW and RRTM referred to twice, and my perception is that all of the acronyms can be removed from the abstract without loss of meaning.

Introduction, p. 25, first sentence: this sounds a little strange "...tend to underestimate the cloud fraction and cloud optical depth due to underestimates in cloud albedo and liquid water path..." as this is not really a cause and effect? One could leave off the phrase beginning with 'due to' with no loss of meaning.

p. 25 line 20-21: actually, close to the coast, the ORACLES campaign also highlights the presence of BBA in the boundary layer during August, e.g., Kacarab et al., 2020, ACP Special Issue. It might be good to qualify this sentence further, either with "initial space-based observations" or, "in September-October".

p. 26, line 25: the Lagrangian approach within Diamond et al2018 was applied to the cloud layer, whereas in the current study, the aerosol layer is followed. These air flows are quite different.

Section 2.1: how are the authors treating orographic effects upon the clouds? And are MFRSR AODs, which come from the elevated AMF1 site, treated the same as the AERONET values taken at the airport? These will have a systematic bias.

p. 28: line 11 mentions MERRA2 thermodynamic profiles are used within the RT calculations, while line 27 indicates the INTERPSONDE profiles are used. Please clarify.

P. 29 top paragraph: the first sentence implies one SSA value was used while line 8 states 3 different SSA values were used. Please clarify.

P. 30 line 14-15: "Brown carbon tends to be more absorbing than organic carbon". Doesn't this depend on how brown carbon is defined? I should also note that the SSA values in Zuidema 2018 correspond to an RH of 45-65%, and to aerosols smaller than 1 micron in aerodynamic diameter, which will exclude the larger sea salt particles possessing a larger SSA.

Section 3 p. 30: Fig. 2: it seems strange to me that BC and OC are not collocated better, e.g., 6 Sept.. What do you make of this? Do BC and OC relate differently to RH within MERRA2 (though this is not the explanation I don't think) ? Section 3 p. 30 Fig. 4: What's also clear about this is that back trajectories associated with higher AODs experience stronger winds. This is why their back trajectories extend further into Africa. Some of this is likely explained by the AEJ-S.

Section 4, Fig. 5: MERRA2 does not seem to capture the boundary layer decoupling at all. It would be worth mentioning how many levels MERRA2 is resolving within the boundary layer, either here or in Section 2.2

p. 31 line 23: are you sure the LASIC radiosondes have been assimilated into MERRA-2? That is quite valuable to know.

p. 31 line 24: it would be helpful to indicate contours for high MERRA-2 values of either BC or BrC (or some combination) on top of the moisture plot.

p. 32 line 14: "exhibit," -> "August 28-29, exhibit"

p. 32 line 18: "-1" needs to be superscripted.

p. 32 line 27: needs to be bolded and enlarged?

Section 5.1, here and in Fig. 9, caption: make clear there is no conditional sampling based on an AOD threshold applied. At least, I believe that is the case.

p. 33 line 17: "-1" needs to be superscripted. This occurs twice.

p. 33 lines 18-19: it would be good to report the RH here, as one would not expect aerosol swelling until a fairly high RH.

p. 33 line 26: the Shinozuka et al. 2020 model-observational comparison paper would be good to cite here as well, as it indicates a wide range of SSAs in use across models.

Section 5;2: several instances of '-2' that need to be superscripted.

p.35 line 24: Does this sentence need to be modified to indicate that it is small aerosols such as BBA that don't impact LW radiation, in contrast to large aerosol such as dust?

Section 5.2, Fig. 12: this indicates DARE is only a cooling, even when cloud is present. I am surprised by this. Does this mean the cloud optical depth or cloud albedo is so small, that a positive DARE can never be actualized? Is DARE being calculated relative to the cloud albedo? Please comment, an additional plot showing the TOA DARE as a function of the underlying albedo might even be a nice addition. If DARE is indeed always a cooling for the MICROBASE cloud properties, that is worth including in the abstract.

p.37 line 21 and line 25: superscript the '-1'

p. 38 line 24: the heating could also help drive an anomalous ascent. Something in fact must happen because the observed temperature profiles don't deviate from each other by say more than 2K, less from day to day when the synoptics are more similar.

p. 38 line 27-28: You could check with Art Sedlacek but I don't think there is evidence for tar balls over the southeast Atlantic.

p. 39, line 19-20: I'm not quite following is, is LW cooling at cloud top reduced because the warmer free-tropospheric temperatures emit more LW, for the same water vapor amount -the effect shown in Fig. 13? Or is it because of the increasing free-tropospheric opacity associated with the moisture colocated with the aerosol, as noted in the correspondence between Fig 2 and 6?

Table 3 caption: superscript '-1'

Overall: latitude and longitude indicators throughout need degree signs.

Check abstract again after you have finished to be sure you feel it captures your most salient findings.

---

## Author Response (AR2)

Dear Authors - Thank you for submitting your manuscript to ACP. I have reviewed the referees' comments and your responses to them, and have a few additional comments I would like you to consider before this manuscript goes to publication. The manuscript will not go back to the referees again. The page and line numbers of the comments are based on the Author's Response file you provided. It doesn't look to me that that shows the tracked changes to the original manuscript, as requested by the journal, but I can live with that. You may follow suit with your next revision but I would like to see your responses indicated point-by-point to the comments below.

Sincerely, Paquita Zuidema

Dear Dr. Zuidema,

Thank you for taking the time to read our manuscript and provide feedback. Our apologies for not including the track changes version in the previous revision. We have track changes in the current version of the manuscript, as well as a response to each of your comments and the comments from Referee #2.

Allie Collow

Abstract, line 18: "this has yet to be quantified" is a bit strong. Radiative heating profiles have been calculated before, but not specifically for the air masses you are targeting. Please consider revising this sentence.

This sentence has been revised: "As the aerosol plume progresses across the southeast Atlantic Ocean, radiative heating within the aerosol layer has the potential to alter the thermodynamic environment and therefore the cloud structure; however, limited work has been done to quantify this along the trajectory of the aerosol plume in the region."

Abstract: My understanding is that acronyms should not be used or defined within the abstract unless substantially used. I only see SW and RRTM referred to twice, and my perception is that all of the acronyms can be removed from the abstract without loss of meaning.

The acronyms have been removed from the abstract. We opted to leave MERRA-2 as an abbreviation since that is the more common form for it.

Introduction, p. 25, first sentence: this sounds a little strange "…tend to underestimate the cloud fraction and cloud optical depth due to underestimates in cloud albedo and liquid water path…" as this is not really a cause and effect? One could leave off the phrase beginning with 'due to' with no loss of meaning.

This sentence has been modified as suggested.

p. 25 line 20-21: actually, close to the coast, the ORACLES campaign also highlights the presence of BBA in the boundary layer during August, e.g., Kacarab et al., 2020, ACP Special Issue. It might be good to qualify this sentence further, either with "initial space-based observations" or, "in September-October".

This sentence has been modified as suggested.

p. 26, line 25: the Lagrangian approach within Diamond et al 2018 was applied to the cloud layer, whereas in the current study, the aerosol layer is followed. These air flows are quite different.

Indeed, the HYSPLIT runs performed by Diamond et al. (2018) originated lower in the atmosphere. The key point we are trying to make in this sentence stems from a much broader conclusion from that paper – a Lagrangian framework is needed. From the abstract, "A Lagrangian framework following the clouds and accounting for the history of smoke entrainment and precipitation is likely necessary for quantitatively studying this system; an Eulerian framework (e.g., instantaneous correlation of A-train satellite observations) is unlikely to capture the true extent of smoke–cloud interaction in the southeast Atlantic."

Section 2.1: how are the authors treating orographic effects upon the clouds? And are MFRSR AODs, which come from the elevated AMF1 site, treated the same as the AERONET values taken at the airport? These will have a systematic bias.

Thanks for bringing this to our attention. The height of the AMF1 site compared to the AERONET site is certainly an issue. After a closer look at the figure, we concluded that the AOD was slightly higher in the AERONET observations and a sentence has been added to the text noting this systematic bias.

p. 28: line 11 mentions MERRA2 thermodynamic profiles are used within the RT calculations, while line 27 indicates the INTERPSONDE profiles are used. Please clarify.

MERRA-2 thermodynamic profiles are only used for the RRTM calculations along the back trajectory. For the calculations over Ascension Island, the INTERPSONDE profiles were used. We have noted this difference in the text.

P. 29 top paragraph: the first sentence implies one SSA value was used while line 8 states 3 different SSA values were used. Please clarify.

This has been clarified to state we also use two additional variations of the MERRA-2 SSA at 550 nm.

P. 30 line 14-15: "Brown carbon tends to be more absorbing than organic carbon". Doesn't this depend on how brown carbon is defined? I should also note that the SSA values in Zuidema 2018 correspond to an RH of 45-65%, and to aerosols smaller than 1 micron in aerodynamic diameter, which will exclude the larger sea salt particles possessing a larger SSA.

Thank you for specifying the SSA values in Zuidema et al. likely do not include sea salt. This is likely an important factor for the overestimation of SSA in MERRA-2. We have included this important point in the text.

Section 3 p. 30: Fig. 2: it seems strange to me that BC and OC are not collocated better, e.g., 6 Sept.. What do you make of this? Do BC and OC relate differently to RH within MERRA2 (though this is not the explanation I don't think) ?

We believe this might be a misinterpretation of the figure. The shading in Figure 2 does not distinguish between black and organic carbon. If this is in reference to the two different panels, the top is 2016 and the bottom is 2017 as noted on the x axis and in the figure caption.

Section 3 p. 30 Fig. 4: What's also clear about this is that back trajectories associated with higher AODs experience stronger winds. This is why their back trajectories extend further into Africa. Some of this is likely explained by the AEJ-S.

This is now noted in the text.

Section 4, Fig. 5: MERRA2 does not seem to capture the boundary layer decoupling at all. It would be worth mentioning how many levels MERRA2 is resolving within the boundary layer, either here or in Section 2.2

It is true that decoupling is not resolved by MERRA-2 and this issue is now noted in the revised text. The boundary layer is poorly resolved in MERRA-2, with only eight vertical levels below the temperature inversion.

p. 31 line 23: are you sure the LASIC radiosondes have been assimilated into MERRA-2? That is quite valuable to know.

Thanks! Upon closer inspection we discovered that the radiosonde data from LASIC were not assimilated into MERRA-2 as we had previously assumed. The closest radiosondes were located at St. Helena. This statement has now been removed from the manuscript.

p. 31 line 24: it would be helpful to indicate contours for high MERRA-2 values of either BC or BrC (or some combination) on top of the moisture plot.

The RH panels of the figure now include contours for AOD as in the heating rate figures.

p. 32 line 14: "exhibit," -> "August 28-29, exhibit"

This has been corrected.

p. 32 line 18: "-1" needs to be superscripted.

It appears the superscripts were not handled properly in the conversion to *.pdf. We are using a different conversion procedure that corrects this problem.

p. 32 line 27: needs to be bolded and enlarged?

The format from the template is used for the section heading.

Section 5.1, here and in Fig. 9, caption: make clear there is no conditional sampling based on an AOD threshold applied. At least, I believe that is the case.

Correct, the contours are there as a guide for the location of the aerosol plume. We have added a sentence to clarify this point.

p. 33 line 17: "-1" needs to be superscripted. This occurs twice.

It appears the superscripts were not handled properly in the conversion to *.pdf. We are using a different conversion procedure that corrects this problem.

p. 33 lines 18-19: it would be good to report the RH here, as one would not expect aerosol swelling until a fairly high RH.

The statement has been removed.  We agree that the RH's are too low to produce significant aerosol swelling.

p. 33 line 26: the Shinozuka et al. 2020 model-observational comparison paper would be good to cite here as well, as it indicates a wide range of SSAs in use across models.

Thanks! The suggested paper is now cited.

Section 5.2: several instances of '-2' that need to be superscripted.

It appears the superscripts were not handled properly in the conversion to *.pdf. We are using a different conversion procedure that corrects this problem.

p.35 line 24: Does this sentence need to be modified to indicate that it is small aerosols such as BBA that don't impact LW radiation, in contrast to large aerosol such as dust?

Yes, and thanks for catching this. This sentence has been made more specific to BBA.

Section 5.2, Fig. 12: this indicates DARE is only a cooling, even when cloud is present. I am surprised by this. Does this mean the cloud optical depth or cloud albedo is so small, that a positive DARE can never be actualized? Is DARE being calculated relative to the cloud albedo? Please comment, an additional plot showing the TOA DARE as a function of the underlying albedo might even be a nice addition. If DARE is indeed always a cooling for the MICROBASE cloud properties, that is worth including in the abstract.

Figure 12 depicts daily mean DARE, however there are instantaneous periods throughout the month of August 2016 when our RRTM calculations indicate a positive DARE at the TOA. These instances occur when the cloud liquid water path and effective radius are elevated (for example on August 28). While we opted not to add an additional figure, we expanded the discussion of the DARE at the TOA in the manuscript and added a sentence to the abstract.

p.37 line 21 and line 25: superscript the '-1'

It appears the superscripts were not handled properly in the conversion to *.pdf. We are using a different conversion procedure that corrects this problem.

p. 38 line 24: the heating could also help drive an anomalous ascent. Something in fact must happen because the observed temperature profiles don't deviate from each other by say more than 2K, less from day to day when the synoptics are more similar.

We agree and this is now noted in the text.

p. 38 line 27-28: You could check with Art Sedlacek but I don't think there is evidence for tar balls over the southeast Atlantic.

Tar balls are no longer mentioned.

p. 39, line 19-20: I'm not quite following this, is LW cooling at cloud top reduced because the warmer free-tropospheric temperatures emit more LW, for the same water vapor amount -the effect shown in Fig. 13? Or is it because of the increasing free-tropospheric opacity associated with the moisture collocated with the aerosol, as noted in the correspondence between Fig 2 and 6?

Warming and moistening are both important modulators of the cloud top LW cooling and commensurate TKE production. To address this comment, we rewrote the final paragraph in the manuscript to specifically reflect how we believe the radiative processes might in the stratocumulus-to-cumulus transition process. We included the potential effects of heating and moistening.

Table 3 caption: superscript '-1'

It appears the superscripts were not handled properly in the conversion to *.pdf. We are using a different conversion procedure that corrects this problem.

Overall: latitude and longitude indicators throughout need degree signs.

Latitude and longitudes within the text now have degree signs.

Check abstract again after you have finished to be sure you feel it captures your most salient findings.

We modified the abstract to better reflect the findings of our study.

Review on revised "Radiative Heating Rate Profiles over the Southeast Atlantic Ocean during the 2016 and 2017 Biomass Burning Seasons" by Collow et al.

The authors have made a substantial effort in revising the manuscript especially the Introduction section, with stronger motivation and literature review. The organization of sections is also better overall. The authors have made thoughtful responses to the reviewer's comments and incorporated most of the suggested changes. I recommend that this version of the manuscript be published pending some minor comments/technical corrections listed below:

Abstract, Line 30: .... extremely sensitive to the single-scattering albedo assumptions in the models.

This sentence has been modified as suggested.

Line 15: biomass burning aerosol plumes/layers extend up to 3- 4.5 km....

This sentence has been modified as suggested.

Line 18-19: 'global models commonly allow' -> please change the wording here to something like, models simulate.

This sentence has been modified as suggested.

Line 33: "SSA are from MERRA-2, and were scaled in the vertical by the profile of mixing ratio for the individual species (GMAO, 2015; GMAO, 2015b). The value for SSA at 550 nm from MERRA-2 was used and assumed to be spectrally independent." Since you mention in your findings that assumptions of SSA are very important for heating rate calculations, could you list a number or a range for SSA values that you calculated in the model for the typical smoke mixture (based on the mixing ratio profiles) you observed in your study? Also, could you add a few sentences on what are the implications on using spectrally 'independent' SSA for you heating rate calculations, and how would they differ from reality?

Monthly mean values for the SSA are now given in section 2.3. We would not expect a large impact on our heating rate calculations by assuming a spectrally independent SSA, which is mentioned in the discussion section. Our basis for this claim is the proximity of 550 nm to the spectral solar maximum irradiance. Specifying the SSA at ~600-660 nm, for example, adds the complication of the ozone absorption, which acts to mitigate any change in SSA by reducing reduce the solar irradiance in some bands as the spectral solar irradiance declines rapidly as wavelength increases. Moving to shorter wavelengths increases the Rayleigh optical thickness, again mitigating changes in SSA. However, we support improving the spectral dependence of the SSA in models as a focus area for researchers in the field.

[revised manuscript text omitted]